# Continual Knowledge Adaptation for Reinforcement Learning

**Jinwu Hu**[1 2*]  **Zihao Lian**[1*]  **Zhiquan Wen**[1*]  **Chenghao Li**[1 2]
**Guohao Chen**[1 2]  **Xutao Wen**[1]  **Bin Xiao**[3†]  **Mingkui Tan**[1 4†]

[1]South China University of Technology, [2]Pazhou Laboratory,
[3]Chongqing University of Posts and Telecommunications
[4]Key Laboratory of Big Data and Intelligent Robot, Ministry of Education,

## Abstract

Reinforcement Learning enables agents to learn optimal behaviors through interactions with environments. However, real-world environments are typically non-stationary, requiring agents to continuously adapt to new tasks and changing conditions. Although Continual Reinforcement Learning facilitates learning across multiple tasks, existing methods often suffer from catastrophic forgetting and inefficient knowledge utilization. To address these challenges, we propose **C**ontinual **K**nowledge **A**daptation for **R**einforcement **L**earning (**CKA-RL**), which enables the accumulation and effective utilization of historical knowledge. Specifically, we introduce a Continual Knowledge Adaptation strategy, which involves maintaining a task-specific knowledge vector pool and dynamically using historical knowledge to adapt the agent to new tasks. This process mitigates catastrophic forgetting and enables efficient knowledge transfer across tasks by preserving and adapting critical model parameters. Additionally, we propose an Adaptive Knowledge Merging mechanism that combines similar knowledge vectors to address scalability challenges, reducing memory requirements while ensuring the retention of essential knowledge. Experiments on three benchmarks demonstrate that the proposed CKA-RL outperforms state-of-the-art methods, achieving an improvement of 4.20% in overall performance and 8.02% in forward transfer. The source code is available at https://github.com/Fhujinwu/CKA-RL.

## 1  Introduction

Reinforcement Learning (RL) has emerged as a powerful paradigm in machine learning, enabling agents to learn optimal behaviors through interactions with dynamic environments [24, 11, 45]. RL has achieved significant success in fields such as robotic control [23, 35], embodied intelligence [7, 15], and natural language processing [17, 10, 18]. However, traditional RL usually assumes a static environment where tasks and data distributions remain fixed, intending to solve a single, well-defined problem [32]. In contrast, real-world environments are typically non-stationary, requiring agents to continuously adapt to new tasks and evolving conditions. In light of this, Continual Reinforcement Learning (CRL) [25] has been introduced to enable agents to adapt and maintain performance across multiple tasks, facilitating more robust decision-making in dynamic environments [14, 2, 50].

*Unfortunately*, CRL still faces several challenges, which are as follows. *1) Cross-task conflict:* In continual learning, tasks may share certain structures or knowledge while also having incompatible goals or constraints. This interplay complicates the direct reuse of previously learned information [27].

---

*Equal contribution. Email: fhujinwu@gmail.com, lianzihaolzh@gmail.com, sewenzhiquan@gmail.com
†Corresponding author. Email: mingkuitan@scut.edu.cn, xiaobin@cqupt.edu.cn

39th Conference on Neural Information Processing Systems (NeurIPS 2025).

*2) Catastrophic forgetting:* When learning new tasks, the agent may overwrite or distort previously acquired knowledge, leading to a loss of performance on earlier tasks [32, 50].

Recently, several attempts have been proposed to enable agents to continuously learn across multiple tasks while maintaining or enhancing their performance on previously acquired tasks [60, 19]. The existing methods can be broadly categorized into four main groups [50]. The *regularization-based methods* [20, 37] introduce regularization terms into the learning objective that penalize large updates to important parameters for previously learned tasks. The *rehearsal-based methods* [9, 59, 58] mitigate forgetting by storing previous experiences in memory and periodically replaying them during the learning of new tasks. By leveraging past experiences, these methods help reduce short-term biases and improve task performance across the sequence. The *architecture-based methods* [49, 32, 5, 39] focus on learning a shared structure, such as modularity or composition, to facilitate continual learning. They reuse parts of previous solutions by forming abstract concepts or skills. The *meta-learning based methods* [40, 12, 30, 8] improve the learning efficiency of agents by utilizing past successes and failures to refine their optimization processes. This creates an inductive bias that enhances sample efficiency and adaptability in acquiring new behaviors.

Although there has been significant progress in existing methods, they still face several limitations as follows. **Firstly**, existing methods, such as PackNet [33], often fail to effectively address task dependencies and conflicts, which makes it challenging to transfer knowledge across tasks without interference or degradation in performance. **Secondly**, most methods still exhibit substantial performance degradation on previously learned tasks upon acquisition of new ones. This issue is particularly pronounced in methods that rely on regularization, as they may struggle to efficiently retain knowledge from earlier tasks as the task sequence lengthens. **Lastly**, many methods face scalability issues, especially as the number of tasks grows significantly, leading to increased memory and computational costs, such as CompoNet [32].

To address these limitations, we propose **C**ontinual **K**nowledge **A**daptation for **R**einforcement **L**earning (**CKA-RL**), which enables the accumulation and effective utilization of historical knowledge, thereby accelerating learning in new tasks and explicitly reducing performance degradation on previous tasks. Specifically, we assume that the agent acquires a unique knowledge vector for each task during continual learning. Based on this, we propose a Continual Knowledge Adaptation strategy, which involves maintaining a task-specific knowledge vector pool and dynamically using historical knowledge to adapt the agent to a new task. This method mitigates catastrophic forgetting and facilitates the efficient transfer of knowledge across tasks by preserving and adapting crucial model parameters. Furthermore, we introduce an Adaptive Knowledge Merging mechanism that clusters and consolidates similar knowledge vectors to address scalability issues, reducing memory requirements while ensuring the retention of essential information.

**Main novelty and contributions. 1)** We propose a novel continual reinforcement learning method, called CKA-RL. It enables agents to reuse knowledge from previously learned tasks, mitigating catastrophic forgetting, and leveraging this historical knowledge to enhance the learning efficiency. **2)** We propose Continual Knowledge Adaptation strategy, which dynamically adapts historical knowledge to a new task. To reduce storage requirements, we introduce an Adaptive Knowledge Merging mechanism that combines similar knowledge vectors, addressing scalability issues. **3)** Experiments demonstrate that the proposed method outperforms state-of-the-art methods on three benchmarks, achieving an improvement of 4.20% in performance and 8.02% in forward transfer.

## 2    Related Work

Continual Reinforcement Learning (CRL) aims to enable agents to continuously learn and optimize strategies in dynamic environments, improving their ability to adapt to environmental changes and achieve goals more efficiently. The existing methods can be broadly categorized into four main types [50]: regularization-based methods, rehearsal-based methods, architecture-based methods, and meta-learning-based methods, each of which is detailed as:

**Regularization-based Methods.** These methods [20, 37] add regularization terms to the training process to balance learning new and old tasks without storing models of old tasks. ITER [20] periodically distills the current policy and value function into a newly initialized network during training and imitates the teacher network using a linear combination of loss terms to enhance model generalization. TRAC [37] adaptively adjusts the regularization strength based on online convex

optimization theory to prevent excessive weight drift, thereby reducing plasticity loss and enabling rapid adaptation to new distribution changes. Anand et al. [3] propose a method that decomposes the value function into permanent and transient components to address the stability-plasticity dilemma in continual reinforcement learning, enabling efficient adaptation and control in dynamic environments.

**Rehearsal-based Methods.** These methods [9, 59, 58, 13] utilize experience replay, generation replay, parameter replay, *etc.*, to store past experiences and prevent catastrophic forgetting in agents. RECALL [58] focuses on improving plasticity and stability in continuous reinforcement learning through multi-head neural network training, coupled with adaptive normalization and policy distillation techniques. IQ [59] employs a context partitioning strategy based on online clustering, combined with multi-head networks and knowledge distillation technology, to reduce interference between different state distributions. DRAGO [13] leverages synthetic experience replay and exploration-based memory recovery to retain knowledge across tasks, mitigating catastrophic forgetting without requiring the storage of past task data.

**Architecture-based Methods.** These methods [49, 32, 5, 39, 34] concentrate on learning a policy with a set of shared parameters to handle all incremental tasks, including parameter allocation, model reorganization, and modular networks. For example, MaskNet [5] employs a learnable modulation mask to isolate the parameters of different tasks on a fixed neural network and accelerates learning new tasks by linearly combining masks from previous tasks. CompoNet [32] uses a modular architecture with self-composing policies to enable efficient knowledge transfer and scalable learning in continual reinforcement learning. COMP [34] decomposes complex tasks into multiple subtasks corresponding to different neural modules. These modules can be combined to form a complete policy. REWIRE [46] redefines the connection methods of the neural network and reorders the neurons in each layer to achieve additional plasticity in unstable environments.

**Meta-Learning based Methods.** These methods [40, 12, 30, 8] assist agents in rapidly adapting to new tasks by simulating the reasoning phase during training. MB-MPO [8] is a model-based method that meta-learns policies robust to ensemble dynamics, reducing reliance on accurate models. MAML [12] is a model-agnostic meta-learning algorithm that explicitly trains models to achieve strong generalization from a small number of gradient updates. ESCP [30] enhances the robustness and responsiveness of context encoding, significantly accelerating adaptation in reinforcement learning involving sudden environmental changes.

## 3 Problem Formulation

The **standard Reinforcement Learning (RL)** problem [47] is modeled as a **Markov Decision Process (MDP)**, with tuple $M = \langle \mathcal{S}, \mathcal{A}, p, r, \gamma \rangle$, where $\mathcal{S}$ is state space, $\mathcal{A}$ is action space, $p : \mathcal{S} \times \mathcal{A} \times \mathcal{S} \to [0, 1]$ is the state transition probability function, $r : \mathcal{S} \times \mathcal{A} \to \mathbb{R}$ is the reward function, and $\gamma \in [0, 1]$ is the discount factor. At each time step $t \in \mathbb{N}$, the agent observes the current state $s_t \in \mathcal{S}$ and takes an action $a_t \sim \pi(\cdot|s_t)$, where $\pi : \mathcal{S} \times \mathcal{A} \to [0, 1]$ is the policy. The environment transitions to a new state $s_{t+1} \sim p(\cdot|s_t, a_t)$, and the agent receives a reward $r(s_t, a_t)$. The goal of RL is to find an optimal policy $\pi^*$ that maximizes the expected discounted return as follows:

$$\max_{\pi} \ \mathbb{E}_{\pi, p}[\sum_{t=0}^{\infty} \gamma^t r(s_t, a_t)], \tag{1}$$

where the expectation is over trajectories generated by policy $\pi$ and state transitions $p$. The discount factor $\gamma$ controls the trade-off between immediate and future rewards.

Given $N$ tasks, the agent is expected to maximize the RL objective for each task in $\mathcal{T} = \{\tau_1, \tau_2, \dots, \tau_N\}$. Each task $\tau_i$ is associated with a distinct MDP $M_i = \langle \mathcal{S}_i, \mathcal{A}_i, p_i, r_i, \gamma_i \rangle$. An RL problem is considered an instance of **Continual Reinforcement Learning (CRL)** if agents are required to continuously learn. Following the definition presented by *Abel et al.* [2], we have the following concept: an RL problem defined by a tuple $(e, \nu, \Lambda)$, where $e$ is the environment, $\nu$ is the performance function, and $\Lambda$ is the set of all agents, the problem is a CRL problem if $\forall_{\lambda^* \in \Lambda^*} \lambda^* \overset{e}{\nrightarrow}$ (never reaches) $\Lambda_B$, where $\Lambda_B \subset \Lambda$ is a basis such that $\Lambda_B \vdash_e$ (generates) $\Lambda$ (in $e$) and $\Lambda^* = \arg\max_{\lambda \in \Lambda} \nu(\lambda, e)$. In other words, the best agents continue to search indefinitely over the basis $\Lambda_B$ and do not converge to a fixed policy. According to our CKA-RL, whenever the number of tasks $|\mathcal{T}|$ increases, particularly when $\tau_{N+1}$ is added, a new knowledge vector $\boldsymbol{v}_{N+1}$ will be introduced. Therefore, the new task will ensure that $\lambda \overset{e}{\nrightarrow} \Lambda_B$, and thus continues to learn.

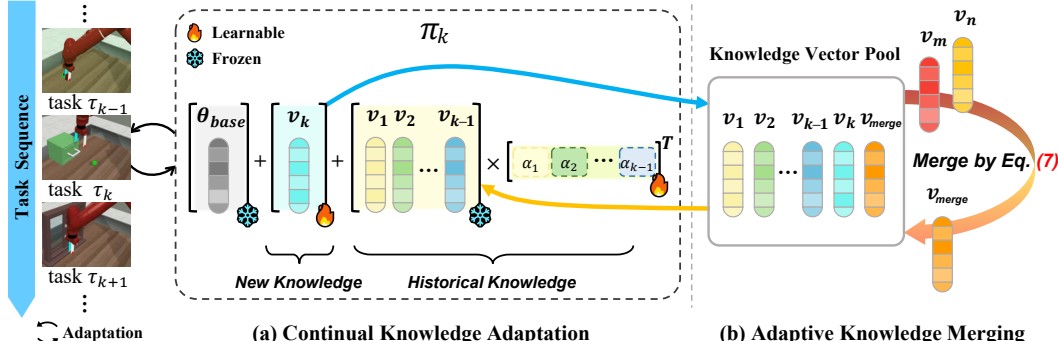

Figure 1: An illustration of the CKA-RL. When learning a new task $\tau_k$, the agent $\pi_k$ adapts by using historical knowledge vectors $\mathcal{V} = \{v_1, \ldots, v_{k-1}\}$ with $|\mathcal{V}| \leq K_{\max}$, and learning a new task-specific knowledge vector $v_k$, while the base parameter $\theta_{\text{base}}$ remains fixed. After training, the new knowledge vector $v_k$ is added to a knowledge vector pool $\mathcal{V}$. To maintain memory efficiency, we merge the most similar pairs of knowledge vectors $(v_m, v_n)$ in $\mathcal{V}$ into $v_{\text{merge}}$ when $|\mathcal{V}| > K_{\max}$, thus ensuring essential knowledge is retained while supporting scalable continual learning across sequential tasks.

## 4   Continual Knowledge Adaptation for Reinforcement Learning

In this paper, we propose **C**ontinual **K**nowledge **A**daptation for **R**einforcement **L**earning (**CKA-RL**), which enables the accumulation and effective utilization of historical knowledge, thereby accelerating learning in new tasks while mitigating catastrophic forgetting. As shown in Figure 1, our proposed CKA-RL is composed of three key components: **1)** *Knowledge Vectors*: These vectors capture task-specific knowledge that is used to adapt the model to new tasks, as detailed in Sec. 4.1. **2)** *Continual Knowledge Adaptation*: This component dynamically uses historical knowledge to adapt the agent to a new task, enabling efficient transfer and retention of knowledge (see in Sec. 4.2). **3)** *Adaptive Knowledge Merging*: This component merges similar knowledge vectors to address scalability issues, reducing memory requirements while ensuring the retention of essential knowledge (see in Sec. 4.3). The pseudo-code of CKA-RL is summarized in Algorithm 1.

### 4.1   Knowledge Vectors

In continual reinforcement learning, leveraging knowledge from previous tasks in dynamic environments is crucial to enhance agent performance in subsequent tasks and mitigate catastrophic forgetting. Therefore, we introduce the concept of knowledge vectors to facilitate continual learning, inspired by model editing techniques [21]. Specifically, the knowledge vector, denoted as $v_k \in \mathbb{R}^d$, represents the learned parameters that adapt the pre-trained model to a specific task $\tau_k$, where $d$ denotes the dimensionality of the model parameter. These vectors can be combined linearly with historical knowledge vectors to enable cross-task knowledge transfer.

Formally, given the pre-trained model weights $\theta_{\text{base}} \in \mathbb{R}^d$, the knowledge vector $v_k$ is optimized during task-specific training to capture incremental adaptations. The final task-specific parameter $\theta_k$ is generated through a combination of the base parameter $\theta_{\text{base}}$, historical knowledge matrix $V_{k-1} = [v_1, v_2, \ldots, v_{k-1}]$, and the current task vector $v_k$, which is as follows:

$$\theta_k = \theta_{\text{base}} + \sum_{i=1}^{k-1} v_i + v_k, \tag{2}$$

A linear combination of multiple knowledge vectors enables the model to reuse knowledge from previous tasks [26, 6]. This highlights the potential of knowledge vectors to facilitate continual knowledge adaptation, effectively tackling key challenges in continual learning.

### 4.2   Continual Knowledge Adaptation

The knowledge vector stores task-specific knowledge, and the knowledge from historical tasks facilitates the transfer of policies to new tasks. Under the Continual RL setting, assuming that the

policy $\pi$ shares a consistent observation space $\mathcal{O}$ and action space $\mathcal{A}$, the policy can be represented using a unified architecture parameterized by $\theta$. To achieve cross-task knowledge transfer, we propose **Continual Knowledge Adaptation** strategy. Specifically, it is first necessary to construct shared base model parameter $\theta_{\text{base}}$. The policy parameter is initialized as $\theta_1$ and trained on the initial task $\tau_1$, with the optimization objective being to maximize the expected discounted return. The optimized parameter $\theta_1$ is then used as the base parameter $\theta_{\text{base}}$, which encompasses general feature representations and serves as the foundation for subsequent knowledge adaptation.

The knowledge from previous tasks is stored as knowledge vectors in a knowledge vector pool $\mathcal{V} = \{v_1, \dots, v_{k-1}\}$, where each $v_i$ represents the knowledge vector for task $\tau_i$. For flexibility, we define the null knowledge as $v_1 = \theta_1 - \theta_{\text{base}} = 0$ and include it in $\mathcal{V}$. When learning a new task $\tau_k$, the model parameter is generated by adapting the historical knowledge vectors in $\mathcal{V}$ with the new task-specific knowledge $v_k$, which is as follows:

$$\theta_k = \theta_{\text{base}} + \sum_{j=1}^{k-1} \alpha_j^k v_j + v_k, \quad \text{where } \sum_{j=1}^{k-1} \alpha_j^k = 1. \tag{3}$$

The $\alpha_k = [\alpha_1^k, \dots, \alpha_{k-1}^k]$ represents the normalized adaptation factors for task $\tau_k$, where $\sum_{j=1}^{k-1} \alpha_j^k = 1$. These factors are derived from learnable parameter $\beta_k = [\beta_1^k, \dots, \beta_{k-1}^k]$ through the softmax function:

$$\alpha_j^k = \frac{\exp \beta_j^k}{\sum_{i=1}^{k-1} \exp(\beta_i^k)}. \tag{4}$$

Here, $\beta_k$ controls the contribution of each historical knowledge vector $v_j$ to the new task $\tau_k$, and $v_k$ is the optimizable knowledge vector for the current task. Notably, we initialize $v_k$ to $0$ to allow the model to gradually learn the task-specific knowledge vector. This prevents large initial adjustments, ensuring stable learning without disrupting previously learned tasks. By setting $\alpha_1 = 1$, the model can disregard previously learned knowledge when it is not beneficial for learning new tasks. This design enables continual adaptation of historical knowledge while preserving task-specific knowledge. During the task training, the base parameter $\theta_{\text{base}}$ is fixed to maintain the foundation for knowledge adaptation, and only the adaptation factors $\alpha_k$ (derived from $\beta_k$) and the current task knowledge vector $v_k$ are optimized. Upon the completion of the training process, the obtained knowledge vector $v_k$ is added to the knowledge vector pool $\mathcal{V}$, thereby facilitating efficient knowledge transfer within dynamic environments.

## 4.3 Adaptive Knowledge Merging

Although the proposed Continual Knowledge Adaptation strategy can accelerate the agent learning process and mitigate catastrophic forgetting by utilizing knowledge vectors from previous tasks, maintaining all vectors becomes impractical as the number of stored knowledge vectors increases. Therefore, we propose **Adaptive Knowledge Merging** to address scalability issues while preserving essential knowledge. Specifically, we merge similar knowledge vectors into compact representations, ensuring that the memory footprint remains manageable. The pairwise similarity between knowledge vectors is measured using normalized cosine similarity as follows:

$$S_{ij} = \frac{v_i \cdot v_j}{\|v_i\| \|v_j\|}, \tag{5}$$

where $S_{ij} \in [-1, 1]$ measures the directional alignment between vectors $v_i$ and $v_j$. A value of $S_{ij} = -1$ indicates that the two knowledge vectors are in direct conflict. Conversely, a value closer to 1 signifies strong alignment, suggesting consistency in the knowledge they encode. This similarity metric effectively captures the functional similarities in how vectors modify the base policy. When the number of knowledge vectors exceeds the maximum capacity $K_{\text{max}}$, which serves as a hyperparameter for controlling memory usage, we compute the pairwise similarity matrix $S$ across all stored knowledge vectors. Then, we identify the most similar pair $(v_m, v_n)$ by finding the pair with the highest similarity score, which is as follows:

$$(v_m, v_n) = \arg\max_{i,j} S_{i,j}. \tag{6}$$

These vectors $v_m$ and $v_n$ are merged by averaging:

$$v_{\text{merge}} = \frac{1}{2}(v_m + v_n), \tag{7}$$

**Algorithm 1** The Pipeline of the Proposed CKA-RL

---

**Input:** Task sequence $\mathcal{T} = \{\tau_1, \ldots, \tau_N\}$, Initial model parameters $\boldsymbol{\theta}_1$, Knowledge pool $\mathcal{V} = \{\mathbf{0}\}$

1: **Base Policy Learning:** $\boldsymbol{\theta}_{\text{base}} \leftarrow \arg\max_{\theta} \mathbb{E}_{\pi_\theta} \left[ \sum_{t=0}^{\infty} \gamma^t r_1(s_t, a_t) \right]$

2: **for** $k \leftarrow 2, \ldots, N$ **do**
3:     **Task Initialization:** $\boldsymbol{v}_k \leftarrow \mathbf{0}, \boldsymbol{\beta}_k \sim \mathcal{N}(0, 1), \boldsymbol{\alpha}_k \leftarrow \text{softmax}(\boldsymbol{\beta}_k)$
4:     **Policy Construction:** $\boldsymbol{\theta}' \leftarrow \boldsymbol{\theta}_{\text{base}} + \sum_{j=1}^{k-1} \alpha_j^k \boldsymbol{v}_j + \boldsymbol{v}_k$
5:     **Policy Optimization:** $\arg\max_{\boldsymbol{v}_k, \boldsymbol{\beta}_k} \mathbb{E}_{\pi_{\theta'}} \left[ \sum_{t=0}^{\infty} \gamma^t r_k(s_t, a_t) \right]$
6:     **Knowledge Preservation:** $\boldsymbol{\theta}_k \leftarrow \boldsymbol{\theta}', \mathcal{V} \leftarrow \mathcal{V} \cup \{\boldsymbol{v}_k\}$
7:     **if** $|\mathcal{V}| > K_{\max}$ **then**
8:         Compute similarity matrix $S$ using Eq.(5)
9:         $(\boldsymbol{v}_m, \boldsymbol{v}_n) \leftarrow \arg\max_{i,j} S_{i,j}$
10:       Merge $\boldsymbol{v}_m$ and $\boldsymbol{v}_n$ into $\boldsymbol{v}_{\text{merge}}$ using Eq.(7)
11:      Update $\mathcal{V}$: $\mathcal{V} \leftarrow (\mathcal{V} \setminus \{\boldsymbol{v}_m, \boldsymbol{v}_n\}) \cup \{\boldsymbol{v}_{\text{merge}}\}$

**Output:** Learned policies $\{\pi_{\theta_1}, \ldots, \pi_{\theta_N}\}$

---

where $\boldsymbol{v}_{\text{merge}}$ is added to the knowledge vector pool $\mathcal{V}$. The merged vector $\boldsymbol{v}_{\text{merge}}$ replaces $\boldsymbol{v}_m$ and $\boldsymbol{v}_n$, reducing memory requirement while maintaining the knowledge from both vectors. It ensures that the knowledge vector pool remains compact and efficient while preserving knowledge from previous tasks. Through iterative merging, this strategy dynamically maintains a balance between memory efficiency and knowledge retention, addressing scalability issues in dynamic environments.

**Remark:** For a detailed mathematical analysis supporting the performance and stability of the proposed method, please refer to Appendix A.

## 5 Experiments

### 5.1 Experimental Settings

**Benchmarks.** We follow the experimental settings established in prior work [32] and compare CKA-RL with SOTA CRL methods across three distinct dynamic task sequences, including 1) Meta-World [57], 2) Freeway [31], and 3) SpaceInvaders [31]. These sequences are designed to evaluate the robustness and generalization capabilities of different methods under varying levels of task complexity and action space characteristics. More details can be seen in Appendix C.

**Metrics.** Following standard evaluation metrics in CRL [55], we report key metrics, including average performance and forward transfer. The **average performance** at step $t$, denoted as $P(t)$:

$$P(t) = \frac{1}{N} \sum_{i=1}^{N} p_i(t). \tag{8}$$

where $p_i(t) \in [0, 1]$ is the success rate on task $i$ at step $t$, and each of the $N$ tasks is trained for $\Delta$ steps, where $N$ is the number of tasks, so the total number of steps is $T = N \cdot \Delta$. Its final value $P(T)$, serving as a conventional evaluation metric in CRL [32, 53], effectively captures the model's stable performance across dynamic environments. The **forward transfer** measures the extent to which a CRL method is able to transfer knowledge across tasks. It is computed as the normalized area between the training curve of the measured run and the training curve of a reference model trained from scratch. Let $p_i^b \in [0, 1]$ denote the reference performance. The forward transfer on task $i$, denoted as $FT_i$, is defined as follows:

$$FT_i = \frac{AUC_i - AUC_i^b}{1 - AUC_i^b}, AUC_i = \frac{1}{\Delta} \int_{(i-1)\cdot\Delta}^{i\cdot\Delta} p_i(t)\, dt, AUC_i^b = \frac{1}{\Delta} \int_0^{\Delta} p_i^b(t)\, dt. \tag{9}$$

The average forward transfer for all tasks $FT$ is defined as:

$$FT = \frac{1}{N} \sum_{i=1}^{N} FT_i. \tag{10}$$

Table 1: Experimental results comparing the proposed method with nine SOTA methods across Meta-World, SpaceInvaders, and Freeway environments. We report the results for average performance (PERF.) and forward transfer (FWT.), with the best results highlighted in **bold**. The proposed CKA-RL achieves superior performance and forward transfer in all three sequences.

| Method | Meta-World | | SpaceInvaders | | Freeway | | Average | |
|---|---|---|---|---|---|---|---|---|
| | PERF. | FWT. | PERF. | FWT. | PERF. | FWT. | PERF. | FWT. |
| Baseline | $0.4191_{\pm0.49}$ | $0.0000_{\pm0.00}$ | $0.6314_{\pm0.27}$ | $0.0000_{\pm0.00}$ | $0.1247_{\pm0.24}$ | $0.0000_{\pm0.00}$ | $0.3917_{\pm0.35}$ | $0.0000_{\pm0.00}$ |
| FT-1 | $0.0313_{\pm0.09}$ | $-0.2142_{\pm0.38}$ | $0.4412_{\pm0.50}$ | $0.6864_{\pm0.25}$ | $0.1512_{\pm0.36}$ | $0.6935_{\pm0.09}$ | $0.2079_{\pm0.36}$ | $0.3886_{\pm0.27}$ |
| FT-N | $0.3774_{\pm0.48}$ | $-0.2142_{\pm0.38}$ | $0.9785_{\pm0.04}$ | $0.6864_{\pm0.32}$ | $0.7532_{\pm0.16}$ | $0.6935_{\pm0.15}$ | $0.7030_{\pm0.29}$ | $0.3886_{\pm0.30}$ |
| ProgNet | $0.4157_{\pm0.49}$ | $-0.0379_{\pm0.13}$ | $0.3757_{\pm0.27}$ | $-0.0075_{\pm0.22}$ | $0.3125_{\pm0.27}$ | $0.1938_{\pm0.31}$ | $0.3680_{\pm0.36}$ | $0.0495_{\pm0.23}$ |
| PackNet | $0.2523_{\pm0.40}$ | $-0.6721_{\pm1.40}$ | $0.2299_{\pm0.30}$ | $-0.0750_{\pm0.13}$ | $0.2767_{\pm0.36}$ | $0.1970_{\pm0.32}$ | $0.2530_{\pm0.36}$ | $-0.1834_{\pm0.83}$ |
| MaskNet | $0.3263_{\pm0.47}$ | $-0.3695_{\pm0.45}$ | $0.0000_{\pm0.00}$ | $-0.3866_{\pm0.53}$ | $0.0644_{\pm0.17}$ | $-0.0503_{\pm0.12}$ | $0.1302_{\pm0.29}$ | $-0.2688_{\pm0.41}$ |
| CReLUs | $0.3789_{\pm0.47}$ | $-0.0089_{\pm0.24}$ | $0.8873_{\pm0.10}$ | $0.5308_{\pm0.29}$ | $0.7835_{\pm0.13}$ | $0.7303_{\pm0.12}$ | $0.6832_{\pm0.29}$ | $0.4174_{\pm0.23}$ |
| CompoNet | $0.4131_{\pm0.50}$ | $-0.0055_{\pm0.20}$ | $0.9828_{\pm0.02}$ | $0.6963_{\pm0.32}$ | $0.7629_{\pm0.12}$ | $0.7115_{\pm0.10}$ | $0.7196_{\pm0.30}$ | $0.4674_{\pm0.23}$ |
| CbpNet | $0.4368_{\pm0.50}$ | $-0.0826_{\pm0.22}$ | $0.8392_{\pm0.11}$ | $0.4844_{\pm0.28}$ | $0.7678_{\pm0.10}$ | $0.7201_{\pm0.07}$ | $0.6813_{\pm0.30}$ | $0.3740_{\pm0.21}$ |
| **CKA-RL** | $\mathbf{0.4642_{\pm0.50}}$ | $\mathbf{-0.0032_{\pm0.21}}$ | $\mathbf{0.9928_{\pm0.01}}$ | $\mathbf{0.7749_{\pm0.20}}$ | $\mathbf{0.7923_{\pm0.10}}$ | $\mathbf{0.7429_{\pm0.07}}$ | $\mathbf{0.7498_{\pm0.29}}$ | $\mathbf{0.5049_{\pm0.17}}$ |

**Methods.** We compare the CKA-RL with nine SOTA methods. 1) Baseline. 2) FT-1 (Fine-Tuning Single Model) [53]. 3) FT-N (Fine-Tuning with Model Preservation) [53]. 4) ProgNet [39]. 5) PackNet [33]. 6) MaskNet [5]. 7) CReLUs [1]. 8) CompoNet [32]. 9) CbpNet [11].

**Implementation Details.** We follow the prior work [32], employing SAC [16] for Meta-World and PPO [42] for Freeway and SpaceInvaders. For high-dimensional Atari inputs (210×160 RGB), a CNN encoder maps images to compact latent features. All tasks are trained for $\Delta = 1M$ steps. We use Adam (momentum $0.9$, second moment $0.999$), with batch sizes $1024/128$ and learning rates $2.5 \times 10^{-4}/1 \times 10^{-3}$ for PPO/SAC. The discount factor is $\gamma = 0.99$. For SAC, the action standard deviation is constrained to $[e^{-20}, e^2]$, with target smoothing coefficient $5 \times 10^{-3}$, auto-tuned entropy coefficient $0.2$, and action noise clipped to $0.5$. Learning starts after $5 \times 10^3$ steps using $10^4$ random actions for exploration. Policy and target networks are updated every $2$ and $1$ steps, respectively, using 3-layer MLPs with 256 hidden units. For PPO, we apply GAE with $\lambda = 0.95$ across 8 parallel environments, gradient clipping at $0.5$, PPO clip of $0.2$, entropy coefficient $0.01$, and 128 rollout steps. The agent uses a 2-layer MLP with 512 units, and advantage normalization is employed. Following [54, 32, 55], CRL is applied only to the actor, while the critic is reinitialized at each task. We conduct experiments using 10 different random seeds to ensure the robustness and reliability of the results.

## 5.2 Comparison Experiments

We compare the proposed CKA-RL with SOTA methods to demonstrate its superior performance and knowledge transfer capability. Experiments are conducted on three distinct task sequences, including Meta-World, Freeway, and SpaceInvaders, as summarized in Table 1 and Figure 2.

**Consistent superiority of CKA-RL across diverse environments.** The comparative results across three environments, as shown in Table 1, demonstrate the exceptional performance of CKA-RL. Notably, the CKA-RL consistently surpasses SOTA CRL methods in all environments. Specifically, when averaged across three environments, the proposed CKA-RL achieves an improvement of at least 4.20% in performance ($0.7196 \rightarrow 0.7498$) and 8.02% in forward transfer ($0.4674 \rightarrow 0.5049$) compared to CompoNet,

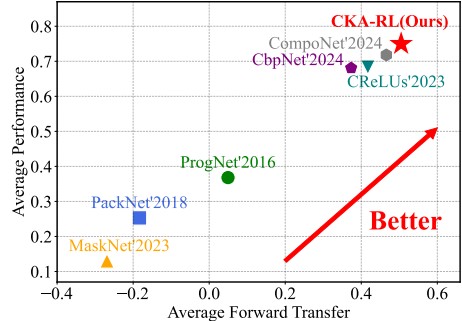

Figure 2: Comparison of SOTA methods with the proposed CKA-RL in terms of average forward transfer and average performance.

demonstrating its superior capability in handling diverse task sequences.

**Robust performance in dynamic environments.** As shown in Table 1, our CKA-RL maintains consistent superiority over existing CRL methods across various dynamic environments. In Meta-

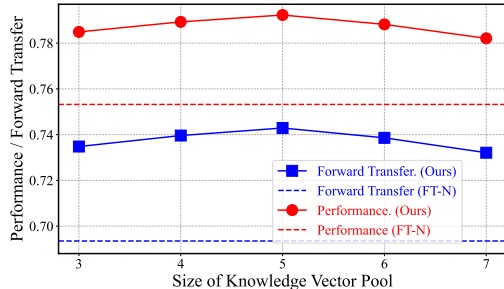
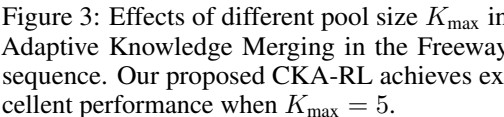
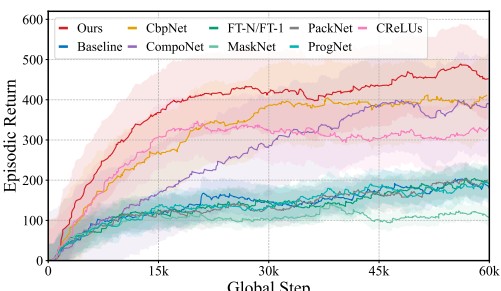

Figure 3: Effects of different pool size $K_{\max}$ in Adaptive Knowledge Merging in the Freeway sequence. Our proposed CKA-RL achieves excellent performance when $K_{\max} = 5$.

Figure 4: Reward curve comparison in the SpaceInvaders sequence. CKA-RL attains a higher initial reward and converges faster, underscoring superior use of historical vectors.

World sequence, CKA-RL achieves a significant performance improvement of 6.27% compared to CbpNet (0.4368 → 0.4642). Similarly, it achieves a performance improvement of 1.02% over CompoNet (0.9828 → 0.9928) in SpaceInvaders sequence and a performance improvement of 1.12% over CReLUs (0.7835 → 0.7923) in Freeway sequence. These consistent improvements across different environments highlight the robustness and stability of our method in dynamic environments.

**Superior knowledge transfer capability.** From Table 1, our CKA-RL consistently achieves superior forward transfer performance across all three environments, demonstrating its exceptional knowledge transfer ability. In Meta-World sequence, CKA-RL shows an improvement of 41.82% compared to CompoNet. Similarly, it outperforms CompoNet (0.6963 → 0.7749) by 11.29% in SpaceInvaders sequence and surpasses CReLUs by 1.73% in Freeway sequence. These results substantiate that our method enables more efficient knowledge transfer compared to existing CRL methods.

**Superior model plasticity.** As shown in Table 1, our method outperforms SOTA plasticity enhancement techniques [11, 1], across all evaluation metrics, including CbpNet and CReLUs. Notably, in SpaceInvaders sequence, our method achieves a significant improvement of 18.30% in average performance (0.8392 → 0.9928) and an 59.97% increase in forward transfer (0.4844 → 0.7749) compared to CbpNet. These comprehensive experimental results demonstrate that our approach surpasses existing plasticity enhancement methods, by effectively leveraging historical knowledge for adaptation. This highlights the potential of adaptive processing to improve model plasticity.

## 5.3 Ablation Studies

**Effectiveness of Components in CKA-RL.** Our CKA-RL enhances the learning process by effectively using historical knowledge vectors and merging redundant vectors. We ablate both components in Table 2. Compared with the base method (FT-N), simply averaging knowledge vectors without adaptive weighting actually performs worse than the baseline (0.7437), indicating that naive merging without adaptation harms performance. In contrast, introducing the Continual Knowledge Adaptation strategy achieves higher performance

Table 2: Effectiveness of components in CKA-RL in Freeway task sequence. 'Avg' is averaging knowledge vectors, 'Adapt' is the Continual Knowledge Adaptation, and 'Merge' is the Adaptive Knowledge Merging.

| Method | Extra Memory | PERF. | FWT. |
|---|---|---|---|
| Base | 0 | 0.7532 | 0.6935 |
| Base + Avg | N | 0.7437 | 0.7127 |
| Base + Adapt | N | 0.7821 | 0.7321 |
| Base + Adapt + Merge | 5 | **0.7923** | **0.7429** |

and forward transfer (*e.g.*, 0.7532 (0.6935) *vs.* 0.7821 (0.7321) on Freeway). This confirms the effectiveness of utilizing historical knowledge vectors for enhancing knowledge transfer across dynamic environments. When further merging redundant knowledge vectors that are similar, our method achieves comparable performance (*e.g.*, 0.7821 → 0.7923 on Freeway), improves the forward transfer (*e.g.*, 0.7321 → 0.7429), and reduces memory requirements (*e.g.*, $N$ → 5), demonstrating the effectiveness of Adaptive Knowledge Merging strategy in addressing scalability issues and maintaining the stable performance on dynamic environments.

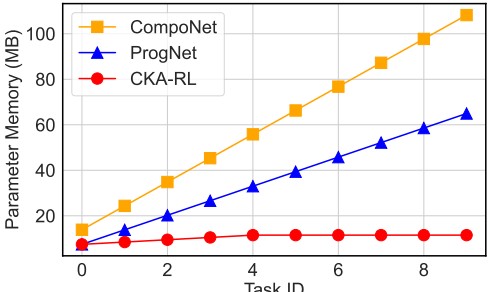
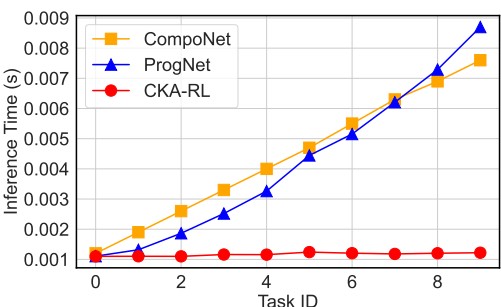

(a) Total parameter memory consumption across an increasing number of sequential tasks.

(b) Inference latency across an increasing number of sequential tasks in SpaceInvaders environment.

Figure 5: Analysis of Total Parameter Memory Consumption and Inference Latency: (a) demonstrates that CKA-RL maintains nearly constant parameter and activation size through vector merging, while others grow linearly; (b) shows that CKA-RL maintains nearly constant inference latency, while others suffer from increasing computational overhead. More details can be found in Appendix E.

**Effects of Pool Size in Adaptive Knowledge Merging.** The size of the knowledge vector pool plays a crucial role in controlling the preservation of historical knowledge. We conduct experiments with $K_{\max}$ values set to $\{3, 4, 5, 6, 7\}$. From Figure 3, our CKA-RL achieves excellent performance when $K_{\max}$ equals 5. Either a smaller or larger $K_{\max}$ hampers the performance. The reasons are as follows. When $K_{\max}$ is small, CKA-RL removes too many knowledge vectors during adaptive knowledge merging, thus being unable to utilize enough knowledge. When $K_{\max}$ is too large, redundant or even conflicting knowledge may slow down the knowledge adaptation process, resulting in performance degradation. Since a larger $K_{\max}$ leads to higher memory requirements, we set $K_{\max}$ to 5 for Freeway and SpaceInvaders experiments, and to 8 for Meta-World, which has a larger number of tasks.

## 5.4 More Discussions

**Superior Initial Performance and Faster Convergence.** One of the key advantages of our proposed CKA-RL is its superior initial performance and rapid convergence, as shown in Figure 4. The reward curve of our CKA-RL demonstrates a higher starting point compared to existing CRL methods, indicating that our method effectively utilizes the historical knowledge vectors. Furthermore, the curve exhibits a steeper ascent, reaching convergence much faster than the competing methods. This not only highlights the efficiency of our CKA-RL but also underscores its effectiveness in utilizing historical knowledge vectors.

**Cross-task Performance of CKA-RL.** To assess how well the final policy consolidates information acquired throughout the learning process, we freeze the model after the last task and evaluate its performance on all previously seen tasks, reporting the average performance across three environments in Table 3. CKA-RL achieves the highest average performance (0.3966), slightly surpassing the closest competitor CbpNet (0.3933) and clearly outperforming FT-N (0.3733) and CReLUs (0.3666). These results indicate that CKA-RL, which combines knowledge adaptation with adaptive merging, results in a more effective final policy that performs better across multiple tasks.

Table 3: Cross-task performance comparison evaluated with the final policy $\pi_{\theta_N}$.

| Method | Pub.'Year | Average Performance |
|---|---|---|
| Baseline | – | 0.2833 |
| FT-N [53] | – | 0.3733 |
| ProgNet [39] | – | 0.2966 |
| PackNet [33] | CVPR'2018 | 0.1900 |
| MaskNet [5] | TMLR'2023 | 0.2266 |
| CReLUs [1] | CoLLAs'2023 | 0.3666 |
| CompoNet [32] | ICML'2024 | 0.3900 |
| CbpNet [11] | Nature'2024 | 0.3933 |
| **CKA-RL (Ours)** | – | **0.3966** |

**Memory Efficiency of CKA-RL.** We evaluate the memory efficiency of CKA-RL by comparing the growth of model parameters across tasks from SpaceInvaders. As shown in Figure 5a, the total

parameter memory of CKA-RL remains nearly constant beyond the fifth task due to our use of the adaptive knowledge merging, which maintains a bounded knowledge vector pool. This ensures that model complexity does not scale linearly with the number of tasks.

**Inference Efficiency of CKA-RL.** From Figure 5b, CKA-RL achieves the highest performance and forward transfer with an average inference time of only **0.0012s** per input. Notably, CKA-RL latency remains essentially constant regardless of the number of tasks, since it consolidates historical knowledge into a fixed-size policy parameter during the parameter construction phase, eliminating the need for complex runtime composition. In contrast, existing architectures such as CompoNet and ProgNet exhibit significant increases in inference cost as the number of tasks grows.

**Effectiveness of CKA-RL in Mitigating Catastrophic Forgetting.** We use the standard forgetting metric [58], which evaluates performance degradation after training on new tasks. As shown in Table 4, CKA-RL achieves a relatively low forgetting rate (0.45), demonstrating its effective retention of previously learned knowledge while continuing to adapt to new tasks. In contrast, PackNet, which shows the lowest forgetting rate (0.04), sacrifices significant overall performance (0.2767). In contrast, CKA-RL achieves the highest performance (0.7923) and forward transfer (0.7429)

Table 4: Forgetting analysis on Freeway. We report overall performance (PERF.), forward transfer (FWT.), and forgetting, computed as pre–post performance on each prior task after learning a new task (lower is better).

| Method | Pub.'Year | PERF. | FWT. | Forgetting |
|---|---|---|---|---|
| ProgNet [39] | – | 0.3125 | 01938 | 0.36 |
| PackNet [33] | CVPR'2018 | 0.2767 | 0.1970 | **0.04** |
| MaskNet [5] | TMLR'2023 | 0.0644 | -0.0503 | **0.04** |
| CReLUs [1] | CoLLAs'2023 | 0.7835 | 0.7303 | 0.52 |
| CompoNet [32] | ICML'2024 | 0.7629 | 0.7115 | 0.49 |
| CbpNet [11] | Nature'2024 | 0.7678 | 0.7201 | 0.46 |
| **CKA-RL (Ours)** | – | **0.7923** | **0.7429** | 0.45 |

across all methods, with a balanced forgetting rate, highlighting its robustness in preventing catastrophic forgetting while maintaining strong performance.

# 6 Conclusion

In this paper, we propose **C**ontinual **K**nowledge **A**daptation for **R**einforcement **L**earning (**CKA-RL**), which enables the accumulation and effective utilization of historical knowledge, thereby accelerating learning in new tasks and explicitly reducing performance degradation on previous tasks. Specifically, we assume that the agent acquires a unique knowledge vector for each task during continual learning. Based on this, we develop a Continual Knowledge Adaptation strategy that enhances knowledge transfer from previously learned tasks. Furthermore, we introduce an Adaptive Knowledge Merging mechanism that combines similar knowledge vectors to address scalability challenges. The CKA-RL outperforms SOTA methods, with a 4.20% overall gain and an 8.02% boost in forward transfer.

# Acknowledgements

This work was partially supported by the Joint Funds of the National Natural Science Foundation of China (Grant No.U24A20327, No.U23B2013).

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

# Supplementary Materials for
# "Continual Knowledge Adaptation for Reinforcement Learning"

## Contents

# A  Mathematical Analysis

We provide some theoretical analysis to support the improved performance of our proposed method. The analysis includes several key lemmas and corollaries, which demonstrate the stability and performance of the knowledge merging mechanism in the context of continual reinforcement learning.

**Preliminaries.** We parameterize the policy for task $\tau_k$ as:

$$\theta_k = \theta_{\text{base}} + \sum_{j=1}^{k-1} \alpha_j^k v_j + v_k, \quad \text{with} \sum_{j=1}^{k-1} \alpha_j^k = 1, \tag{11}$$

where the normalized adaptation factors $\alpha^k$ are produced by a softmax over learnable $\beta^k$ (Eq. (3)–(4)). When the pool size exceeds $K_{\max}$, we merge the most similar knowledge vectors by averaging (Eq. (5)–(7)).

**Lemma 1 (Drift bound under convex reuse).** Let $\Delta_k := \theta_k - \theta_{\text{base}} = \sum_{j<k} \alpha_j^k v_j + v_k$. Since $\alpha^k$ are nonnegative and sum to 1, we have:

$$\|\Delta_k\| \leq \left\| \sum_{j<k} \alpha_j^k v_j \right\| + \|v_k\| \leq \sum_{j<k} \alpha_j^k \|v_j\| + \|v_k\| \leq \max_{j<k} \|v_j\| + \|v_k\|. \tag{12}$$

Thus, the deviation from the base is controlled by the magnitudes of (a small subset of) vectors actually reused and the current task vector. This uses the normalization of $\alpha^k$ given by Eq. (3)–(4).

**Corollary 1 (Lipschitz Performance Stability).** If the task-$k$ return $J_k(\theta)$ is $L$-Lipschitz in parameters, then:

$$|J_k(\theta_k) - J_k(\theta_{\text{base}} + v_k)| \leq L \left\| \sum_{j<k} \alpha_j^k v_j \right\| \leq L \sum_{j<k} \alpha_j^k \|v_j\|. \tag{13}$$

Thus, reusing historical vectors cannot hurt beyond a tunable, data-dependent bound, and the bound tightens as $\alpha^k$ concentrates on small-norm or well-aligned vectors (see Eq. (3)–(4)).

**Lemma 2 (Interference Reduces with Near-Orthogonality).** Let $S_{ij} = \frac{v_i^\top v_j}{\|v_i\|\|v_j\|}$ be the cosine similarity. If $|S_{ij}| \leq \varepsilon \ll 1$ for $i \neq j$, then:

$$\left\| \sum_{j<k} \alpha_j^k v_j \right\|^2 = \sum_{j<k} (\alpha_j^k)^2 \|v_j\|^2 + \sum_{i \neq j} \alpha_i^k \alpha_j^k \|v_i\| \, \|v_j\| S_{ij} \leq \sum_{j<k} (\alpha_j^k)^2 \|v_j\|^2 + \varepsilon \sum_{i \neq j} \alpha_i^k \alpha_j^k \|v_i\| \, \|v_j\|. \tag{14}$$

Hence, the "cross-task" term is $O(\varepsilon)$. Our empirical cosine analysis shows knowledge vectors are nearly orthogonal, with off-diagonal values in $[-0.24, 0.12]$, while full fine-tuned parameters have strong correlations ($0.93 \sim 0.95$). This matches the design goal of reducing interference.

**Corollary 2 (Combining Lemma 1 and Lemma 2).** Combining Lemma 1 and Lemma 2, both the drift and the cross-terms that cause interference are controlled—explaining the improved retention seen in the final-policy evaluation (Table 3).

**Lemma 3 (Bounded Error of Adaptive Merging).** Suppose we must replace $(v_m, v_n)$ by $v_{\text{merge}} = \frac{1}{2}(v_m + v_n)$ when $|V| > K_{\max}$. For any convex coefficients $\lambda, \mu \geq 0, \lambda + \mu = 1$, we have:

$$\left\| \lambda v_m + \mu v_n - v_{\text{merge}} \right\| \leq \tfrac{1}{2} \|v_m - v_n\|. \tag{15}$$

If their cosine similarity $S_{mn} \geq 1 - \delta$ with $\delta \in [0, 2]$, then:

$$\|v_m - v_n\| \leq \sqrt{2(1 - S_{mn})} \max(\|v_m\|, \|v_n\|) \leq \sqrt{2\delta} \max(\|v_m\|, \|v_n\|). \tag{16}$$

Thus, the parameter perturbation induced by merging is $O(\sqrt{\delta})$.

**Corollary 3 (Performance Stability under Merging).** With $L$-Lipschitz $J_k$, we have:

$$|J_k(\theta^{\text{after-merge}}) - J_k(\theta^{\text{before-merge}})| \leq \tfrac{L}{2} \|v_m - v_n\| \leq \tfrac{L}{2} \sqrt{2(1 - S_{mn})} \max(\|v_m\|, \|v_n\|). \tag{17}$$

Hence, merging similar vectors (large $S_{mn}$) has a small, explicitly bounded effect, justifying our "merge-the-most-similar" rule in Eq. (6)–(7).

# B  More Related Work

Reinforcement Learning (RL) [24, 29] constitutes a paradigm within machine learning wherein an agent learns to optimize its decision-making process through interaction with an environment. This interaction involves performing actions and receiving consequent feedback, typically in the form of rewards or penalties. The principal learning objective in RL is the maximization of a cumulative reward signal. In contrast to supervised learning, which relies on datasets comprising pre-defined input-output pairs for model training, RL entails an agent acquiring knowledge from the repercussions of its actions, mediated by this reward-penalty mechanism. This iterative, trial-and-error learning process, coupled with its emphasis on sequential decision-making under uncertainty, distinguishes RL from supervised learning methodologies that depend on labeled datasets. Existing reinforcement learning algorithms can be broadly categorized based on whether an explicit model of the environment is learned or utilized, leading to two principal classes: Model-free RL and Model-based RL.

**Model-free RL.** Model-free RL algorithms enable the agent to learn optimal policies directly from trajectory samples accrued through interaction with the environment, without explicitly constructing an environmental model. Within model-free RL, algorithms are further distinguished by the components they learn, leading to three primary sub-categories: actor-only, critic-only, and actor-critic algorithms. Actor-only algorithms directly learn a policy network, denoted as $\pi_\theta(a|s)$, which maps states to actions. This network takes the current state $s_t$ as input and outputs the action $a_t$. Prominent examples of such algorithms include Reinforce [52] and various policy gradient methods [48]. Critic-only algorithms, in contrast, focus solely on learning a value function (e.g., state-value or action-value function). Given a state $s_t$, the learned value model is used to evaluate all possible actions $a' \in A$, and the action $a_t$ yielding the maximum estimated value is selected. This category encompasses methods such as Q-learning [51]. Actor-critic algorithms combine these two approaches by concurrently maintaining and learning both a policy network (the actor) for action selection and a value function model (the critic) for evaluating actions or states. This category includes algorithms such as Deep Deterministic Policy Gradient (DDPG) [28], Trust Region Policy Optimization (TRPO) [41], Proximal Policy Optimization (PPO) [42], and Asynchronous Advantage Actor-Critic (A3C) [36]. Notably, PPO has gained considerable traction for training large language models. Recent advancements in this area include GRPO [43], which employs group-based advantage estimates within a KL-regularized loss function to reduce computational overhead and enhance update stability, and DAPO [56], which utilizes distinct clipping mechanisms and adaptive sampling techniques to improve efficiency and reproducibility during the fine-tuning of large-scale models.

**Model-based RL.** Model-based RL algorithms endeavor to learn an explicit model of the environment, thereby addressing challenges related to sample efficiency. This is because the agent can leverage the learned model for planning and decision-making, reducing the necessity for extensive direct environmental interaction. The learned representation of the environment is commonly termed a 'world model'. This world model typically predicts the subsequent state $s_{t+1}$ and the immediate reward $r_t$ based on the current state $s_t$ and the action $a_t$ taken. Exemplary model-based RL algorithms include Dyna-Q [38], Model-Based Policy Optimization (MBPO) [22], and Adaptation Augmented Model-based Policy Optimization (AMPO) [44].

# C  Environments and Tasks

Our continual learning experiments evaluate agents across three complementary task domains designed to test different capabilities: robotic manipulation with continuous control (Meta-World) [57], and two vision-based Atari environments [31, 4] emphasizing dynamic decision-making (SpaceInvaders) and sparse-reward navigation (Freeway). This combination spans key RL challenges including high-dimensional state spaces, delayed rewards, procedural variations, and partial observability.

The Meta-World tasks evaluate precise motor control and tool manipulation, while the Atari environments provide contrasting challenges. SpaceInvaders tests rapid visual processing and threat response under varying enemy behaviors, and Freeway examines strategic planning in sparse-reward conditions with evolving obstacle patterns. Collectively, these environments form a comprehensive benchmark for evaluating continual learning.

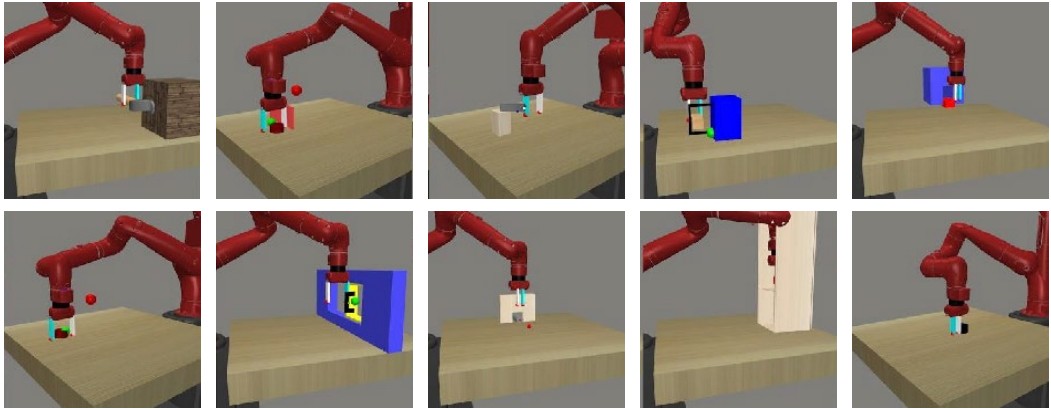

Figure 6: Example frames from each of the 10 Meta-World tasks used in the continual learning benchmark. Tasks are repeated twice to evaluate sequential skill acquisition.

## C.1 Meta-World

The robotic manipulation sequence utilizes the Meta-World benchmark [57], featuring 39-dimensional state observations (encoding arm/object positions) and 4-dimensional continuous actions (arm displacement and gripper torque in $[-1, 1]$). To evaluate continual learning capabilities, we adopt a 20-task sequence consisting of 10 distinct manipulation tasks repeated twice, following the Continual World (CW20) benchmark [53]. This selection covers diverse manipulation skills while maintaining consistent evaluation protocols with prior work. The following lines describe the selected tasks:

**hammer-v2.** Hammer a screw into a wall with randomized initial positions.

**push-wall-v2.** Navigate a puck around obstacles to reach a target location.

**faucet-close-v2.** Rotate a faucet handle clockwise from variable starting positions.

**push-back-v2.** Position a mug beneath a coffee machine with spatial randomization.

**stick-pull-v2.** Retrieve a box using a stick as a tool.

**handle-press-side-v2.** Apply lateral force to press down a handle.

**push-v2.** Basic puck pushing to variable target locations.

**shelf-place-v2.** Precisely place a puck onto a shelf.

**window-close-v2.** Slide a window closed from randomized openings.

**peg-unplug-side-v2.** Remove a laterally mounted peg.

Each episode features randomized object and goal positions, testing robustness to environmental variations. The task sequence progresses from basic manipulations (pushing) to complex tool-use scenarios (hammering, stick-pulling), providing a comprehensive benchmark for evaluating generalization across skills.

## C.2 SpaceInvaders

This arcade-style challenge utilizes the *ALE/SpaceInvaders-v5* [3] environment from the Arcade Learning Environment. In this classic game, the agent controls a laser cannon to defend Earth against descending alien invaders. Observations are provided as RGB frames (210×160×3), with the action space comprising six discrete actions: NOOP (no operation), FIRE, RIGHT, LEFT, RIGHTFIRE (combined movement and firing), and LEFTFIRE. The agent has three lives, and the game terminates when either all lives are lost or invaders reach the ground. Rewards are granted for destroying invaders, with higher-value targets located in back rows.

---

[3]Additional details are available at `https://ale.farama.org/environments/space_invaders/`.

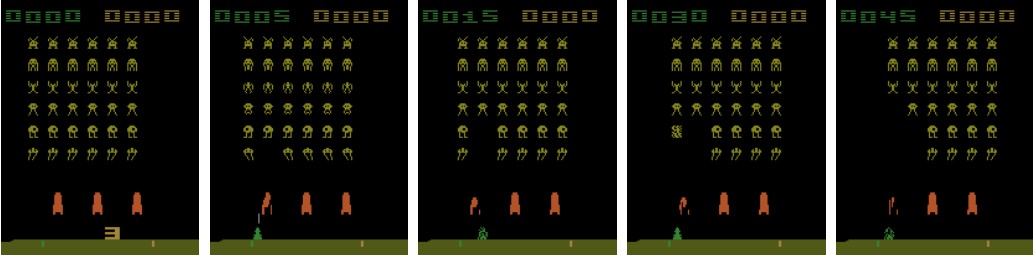

Figure 7: Example gameplay frames from the SpaceInvaders environment.

To systematically evaluate the agent's capability in dynamic threat scenarios, we examine ten strategically selected game modes that modify enemy behavior and environmental dynamics:

**Mode 0 (Baseline).** Standard configuration with static shields and predictable bomb trajectories.

**Mode 1 (Mobile Shields).** Shields oscillate horizontally, eliminating reliable cover positions.

**Mode 2 (Zigzag Bombs).** Invader bombs follow non-linear trajectories, increasing evasion difficulty.

**Mode 3 (Composite Challenge).** Combines mobile shields (Mode 1) with zigzag bombs (Mode 2).

**Mode 4 (High-Speed Bombs).** Baseline configuration with accelerated bomb descent rates.

**Mode 5 (Mobile Shields + Fast Bombs).** Integrates Mode 1's dynamic shields with Mode 4's bomb velocity.

**Mode 6 (Zigzag + Fast Bombs).** Combines Mode 2's erratic bomb paths with increased speed.

**Mode 7 (Full Complexity).** Merges all modifiers: mobile shields, zigzag bombs, and high velocity.

**Mode 8 (Intermittent Visibility).** Invaders periodically become invisible, testing memory and prediction.

**Mode 9 (Dynamic Visibility).** Mobile shields (Mode 1) coupled with intermittent invader visibility.

### C.3 Freeway

The Freeway experiments are conducted using the *ALE/Freeway-v5* [4] environment from the Arcade Learning Environment. In this environment, the agent controls a chicken attempting to cross a multi-lane highway with moving vehicles. Observations are provided as RGB frames (210×160×3), with action space consists of three discrete actions: NOOP (no operation), UP (move forward), and DOWN (move backward). Rewards are exceptionally sparse, the agent only receives +1 upon successfully reaching the top of the screen after crossing all traffic lanes.

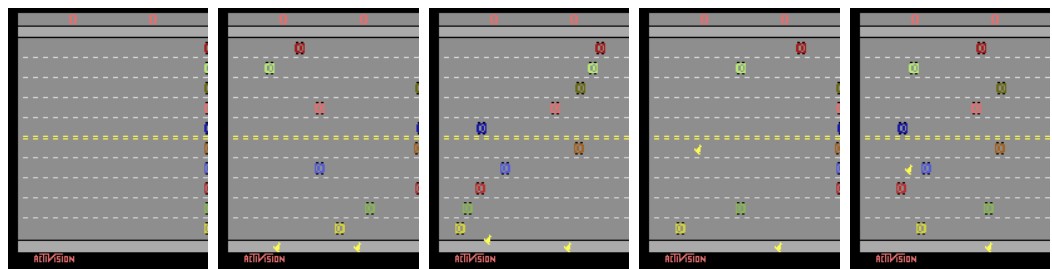

Figure 8: Example gameplay frames from the Freeway environment.

To evaluate the agent's adaptability across varying difficulty levels, we select eight distinct game modes from the available configurations, each modifying traffic patterns and vehicle behaviors:

**Mode 0 (Default).** Default configuration with standard traffic density and vehicle speeds.

---

[4]Additional details are available at `https://ale.farama.org/environments/freeway/`.

**Mode 1 (Increased Traffic & Trucks).** Increased traffic density with faster vehicles. Introduces trucks in the upper lane closest to the center - these longer vehicles require more strategic avoidance.

**Mode 2 (High-Speed Trucks).** Enhanced difficulty from Mode 1 with trucks moving at higher speeds and further increased traffic density.

**Mode 3 (All-Lane Trucks).** Maximum truck presence with trucks appearing in all lanes, maintaining the high speeds established in Mode 2.

**Mode 4 (Randomized Speeds).** Dynamic speed variation where vehicle velocities change randomly during episodes, while maintaining similar traffic density to previous modes without trucks.

**Mode 5 (Clustered Vehicles & Speed Variability).** Combines characteristics of Mode 1 with additional stochastic elements: vehicle speeds vary dynamically and some vehicles appear in tightly-spaced clusters (2-3 vehicles).

**Mode 6 (Heaviest Traffic with Clusters).** Builds upon Mode 5 with the most dense traffic configuration, creating the most challenging navigation scenario.

**Mode 7 (All-Lane Trucks with Random Speeds).** All lanes are filled with trucks, and their speeds vary randomly during the episode.

## D    Implementation Details

To ensure fairness and reproducibility, we build upon the official implementations released by CompoNet [32], which provide well-tested baselines for standard reinforcement learning algorithms. Our modifications to the original SAC and PPO codebases are kept minimal: we only substitute the agent definition with our proposed architecture and introduce the corresponding learning mechanisms.

Both Soft Actor-Critic (SAC) and Proximal Policy Optimization (PPO) follow the standard actor-critic paradigm, where the actor samples actions from a parameterized policy distribution and the critic estimates state values. In all continual reinforcement learning (CRL) methods, the continual adaptation mechanisms are applied only to the actor network, while the critic is reinitialized at the beginning of each task, following common practice in the literature.

Table 5: Hyperparameters shared by all methods in the Meta-World task sequence under the SAC algorithm.

|  | Description | Value |
|---|---|---|
| **Common** | Optimizer | *Adam* |
|  | Adam's $\beta_1$ and $\beta_2$ | (0.9, 0.999) |
|  | Discount rate ($\gamma$) | 0.99 |
|  | Max Std. | $e^2$ |
|  | Min Std. | $e^{-20}$ |
|  | Activation Function | *ReLU* |
|  | Hidden Dimension ($d_{\text{model}}$) | 256 |
| **SAC Specific** | Batch size | 128 |
|  | Buffer size | $10^6$ |
|  | Target Smoothing Coef. ($\tau$) | 0.005 |
|  | Entropy Regularization Coef. ($\alpha$) | 0.2 |
|  | Auto. Tuning of $\alpha$ | Yes |
|  | Policy Update Freq. | 2 |
|  | Target Net. Update Freq. | 1 |
|  | Noise Clip | 0.5 |
|  | Number of Random Actions | $10^4$ |
|  | Timestep to Start Learning | $5 \times 10^3$ |
| **Networks** | Target Net. Layers | 3 |
|  | Critic Net. Layers | 3 |
|  | Actor's Learning Rate | $10^{-3}$ |
|  | Q Networks' Learning Rate | $10^{-3}$ |

For Meta-World tasks, we adopt SAC as the underlying optimization algorithm. Both the actor and critic networks are implemented as two-layer multilayer perceptrons (MLPs), each followed by separate linear output heads predicting the mean and log standard deviation of the Gaussian policy.

For Atari-based tasks including SpaceInvaders and Freeway, PPO is used for training. All methods employ a shared encoder network to extract compact feature representations from image observations. Two single-layer output heads are used to produce the categorical policy logits (actor) and the scalar value estimates (critic). Unless otherwise noted, all hyperparameters are kept identical across different methods and are consistent with those in the reference implementations.

Table 6: Hyperparameters shared by all methods in the SpaceInvaders and Freeway task sequences under the PPO algorithm.

| | Description | Value |
|---|---|---|
| **Common** | Optimizer | Adam |
| | AdamW's $\beta_1$ and $\beta_2$ | (0.9, 0.999) |
| | Max. Gradient Norm | 0.5 |
| | Discount Rate ($\gamma$) | 0.99 |
| | Activation Function | ReLU |
| | Hidden Dimension ($d_{\text{model}}$) | 512 |
| | Learning Rate | $2.5 \cdot 10^{-4}$ |
| **PPO Specific** | PPO Value Function Coef. | 0.5 |
| | GAE $\lambda$ | 0.95 |
| | Num. Parallel Environments | 8 |
| | Batch Size | 1024 |
| | Mini - Batch Size | 256 |
| | Num. Mini - Batches | 4 |
| | Update Epochs | 4 |
| | PPO Clipping Coefficient | 0.2 |
| | PPO Entropy Coefficient | 0.01 |
| | Learn. Rate Annealing | Yes |
| | Clip Value Loss | Yes |
| | Normalize Advantage | Yes |
| | Num. Steps Per Rollout | 128 |

**Methods.** We compare the CKA-RL with nine SOTA methods. **1) Baseline** involves training a randomly initialized neural network for each task, providing a fundamental and essential reference point for comparison. **2) FT-1 (Fine-Tuning Single Model)** [53] continuously fine-tunes a single neural network model across all relevant tasks. **3) FT-N (Fine-Tuning with Model Preservation)** [53] follows a similar fine-tuning approach but maintains separate model instances for each task to mitigate forgetting. **4) ProgNet** [39] instantiates a new neural network whenever the task changes, freezing the parameters of the previous modules and adding lateral connections between their hidden layers. **5) PackNet** [33] stores the parameters to solve every task of the sequence in the same network by building masks to avoid overwriting the ones used to solve previous tasks. **6) MaskNet** [5] leverages previous score parameters across tasks to learn task-specific score parameters, which are then used to generate masks for the neural network, dynamically adapting the model to new tasks. **7) CReLUs** [1] enhances the agent capability in CRL by concatenating ReLUs, which reduces the incidence of zero activations and addresses the plasticity loss issue in neural networks. **8) CompoNet** [32] introduces a scalable neural network architecture that dynamically composes action outputs from previously learned policy modules, rather than relying on shared hidden layer representations across tasks. **9) CbpNet** [11] uses a variation of back-propagation, continually and randomly reinitializing a small fraction of underutilized units to maintain the plasticity of neural networks.

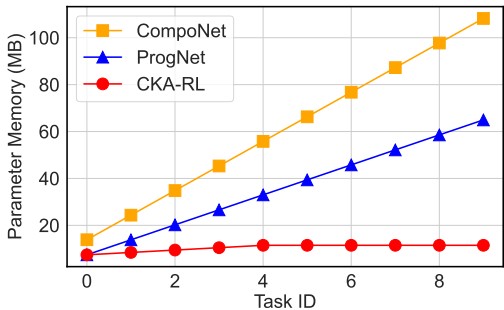
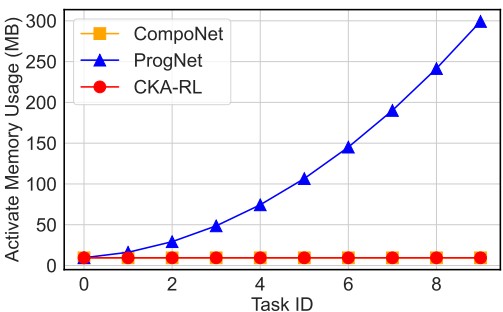

(a) Total parameter memory across tasks. CKA-RL plateaus after 5 tasks, while others grow linearly.

(b) Activation related memory usage during training. CKA-RL remains flat, whereas ProgNet shows quadratic growth.

Figure 9: **Memory cost comparison.** CKA-RL maintains nearly constant parameter and activation size through vector merging, while baselines show linear or even quadratic memory growth.

# E  Efficiency of CKA-RL

## E.1  Performance *vs.* Memory Cost Analysis

We evaluate the memory efficiency of CKA-RL by comparing the growth of model parameters and activation-related memory across tasks from SpaceInvaders. As shown in Figure 9a, the total parameter memory of CKA-RL remains nearly constant beyond the fifth task due to our use of the adaptive knowledge merging, which maintains a bounded knowledge vector pool. This ensures that model complexity does not scale linearly with the number of tasks. In contrast, both ProgNet and CompoNet show linearly increasing memory usage, reaching 64.95 MB and 108.21 MB at the tenth task, respectively.

As shown in Figure 9b, the activation-related memory usage in CKA-RL stays nearly constant throughout training. After merging, the knowledge vectors and base parameter construct a fixed-size policy parameter, which drastically reduces the activation memory overhead. Notably, the memory usage in ProgNet exhibits a quadratic growth trend, whereas CKA-RL remains constant. This suggests severe scalability limitations in ProgNet. This memory overhead in ProgNet primarily stems from its architectural design: during training and inference, it relies on the hidden representations of all previously learned policies. As more tasks are added, the number of such dependencies increases, leading to substantial growth in activation memory consumption.

Table 7: **Performance and Forward Transfer Comparison.** CKA-RL outperforms existing methods across three benchmarks while using significantly less memory.

| Method | Pub.'Year | Meta-World | | SpaceInvaders | | Freeway | | Average | |
|---|---|---|---|---|---|---|---|---|---|
| | | PERF. | FWT. | PERF. | FWT. | PERF. | FWT. | PERF. | FWT. |
| ProgNet [39] | – | 0.4157 | -0.0379 | 0.3757 | -0.0075 | 0.3125 | 0.1938 | 0.3680 | 0.0495 |
| CompoNet [32] | ICML'2024 | 0.4131 | -0.0055 | 0.9828 | 0.6963 | 0.7629 | 0.7115 | 0.7196 | 0.4674 |
| **CKA-RL (Ours)** | – | **0.4642** | **-0.0032** | **0.9928** | **0.7749** | **0.7923** | **0.7429** | **0.7498** | **0.5049** |

CKA-RL achieves a forward transfer of **0.7749** with only **11.49MB** of memory overhead, highlighting its superior efficiency. Unlike methods that accumulate task-specific modules, our approach leverages compact knowledge vectors to consolidate transferable knowledge. This design allows CKA-RL to outperform CompoNet and ProgNet with a significantly smaller memory footprint.

As shown in Table 7, CKA-RL not only achieves the best average performance and forward transfer across diverse tasks (Meta-World, SpaceInvaders, Freeway), but also maintains high efficiency in historical knowledge utilization. These results underscore the practicality of CKA-RL in continual learning scenarios with constrained memory budgets.

## E.2 Performance *vs*. Inference Cost Analysis

We further compare the inference efficiency of different methods on the SpaceInvaders benchmark. As shown in Figure 10, CKA-RL achieves the highest performance and forward transfer with an average inference time of only **0.0012s** per input. Importantly, its inference latency remains almost constant regardless of the number of tasks. This is because CKA-RL consolidates historical knowledge into a fixed-size policy parameter during the parameter construction phase, eliminating the need for complex runtime composition.

In contrast, existing architectures such as CompoNet and ProgNet exhibit significant increases in inference cost as the number of tasks grows. CompoNet shows a linear growth trend in inference time, since its action selection relies on

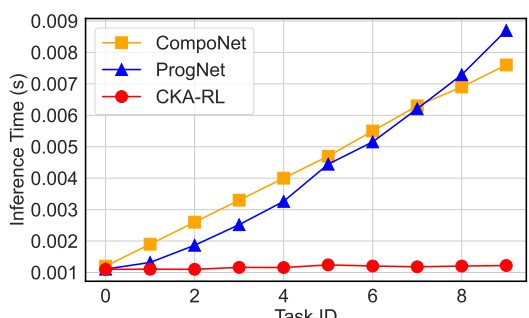

Figure 10: **Inference time *vs*. number of tasks on SpaceInvaders.** CKA-RL maintains nearly constant inference latency due to adaptive knowledge merging, while CompoNet and ProgNet suffer from increasing computational overhead.

aggregating outputs from all previously learned policies. ProgNet, on the other hand, demonstrates a quadratic growth pattern due to its dependence on all intermediate hidden layers for knowledge integration during inference.

These results underscore the scalability advantage of CKA-RL. By offloading the knowledge fusion process to the model-building stage, it minimizes computational overhead at test time, making it well-suited for continual learning in resource-constrained or real-time environments.

# F   Further Experimental Results

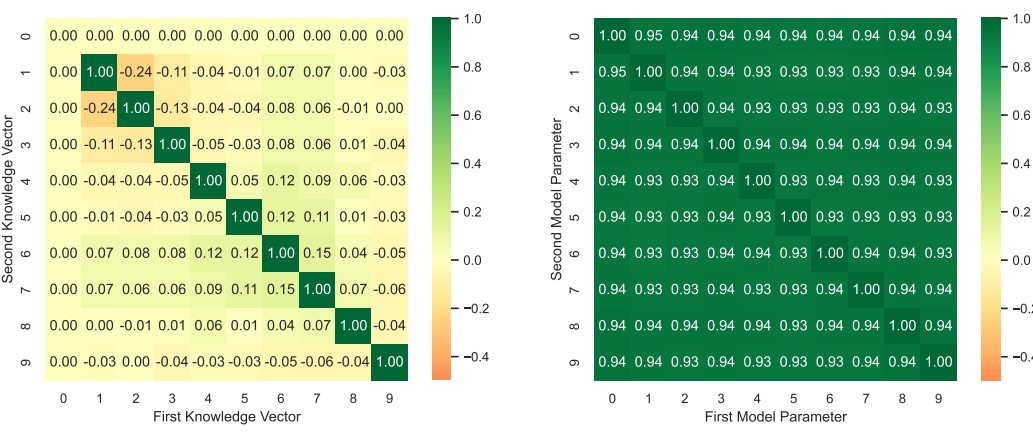

(a) Knowledge vector similarity (from CKA-RL)    (b) Full parameter similarity (from FN)

Figure 11: Cosine similarity of task representations: (a) Knowledge vectors from CKA-RL exhibit near-orthogonal structure across tasks, with off-diagonal values ranging from $-0.24$ to $0.12$; the zero vector for the first task reflects the base model initialization. (b) In contrast, full parameters from standard fine-tuning show strong off-diagonal correlations ($0.93 \sim 0.95$), indicating significant parameter overlap and interference across tasks.

Our CKA-RL decomposes policy parameters into stable base weights $\theta_{\text{base}}$ and task-specific knowledge vectors $\{v_i\}$. To validate this decomposition, we compute pairwise cosine similarities between all parameters of models fine-tuned on different tasks (Figure 11b). The strong off-diagonal correlations ($0.93 \sim 0.95$) reveal that standard sequential fine-tuning causes substantial parameter overlap. Then we analyze the cosine similarities between our learned knowledge vectors $v_i$ (Figure 11a). The near-zero off-diagonal values ($-0.24 \sim 0.12$) demonstrate that knowledge vectors occupy nearly

orthogonal directions in parameter space. The first task's zero-valued vector reflects the initial base model training phase.

This decomposition brings two key advantages: (1) stability, by preserving the base parameters $\theta_{\text{base}}$ across tasks, and (2) plasticity, through task-specific knowledge vectors that remain nearly orthogonal. The orthogonality ensures minimal interference between tasks while enabling effective adaptation. As a result, CKA-RL achieves a forward transfer of 0.7749 and an average performance of 0.9928 across tasks in continual learning settings, outperforming standard fine-tuning (0.6864 and 0.9785, respectively).

# G    Discussions and Future Works

## G.1    Limitations and Future Works

In this paper, we propose **C**ontinual **K**nowledge **A**daptation for **R**einforcement **L**earning (**CKA-RL**), which enables the accumulation and effective utilization of historical knowledge. However, we believe that there are potential studies worth exploring in the future to further capitalize on the advantages of CKA-RL:

- **Complex-Environment Evaluation:** Our current experiments focus on standard continual reinforcement learning benchmarks, and the scalability and robustness of CKA-RL in complex, real-world settings (e.g., high-dimensional visual perception or long-horizon robotic control) remain unverified. In future work, we will extend our evaluation to domains with richer dynamics and observation modalities (e.g., outdoor vision-language navigation tasks) to rigorously assess the generality and practical utility of CKA-RL.

- **Large-Scale Architecture Generalization:** CKA-RL has been validated only on small-scale neural networks, and its applicability to deeper or novel architectures remains untested. In future work, we will evaluate CKA-RL within large-scale models, such as during the RLHF phase of LLMs training, to assess its scalability and effectiveness in deeper networks.

## G.2    Broader Impacts

**Positive Societal Impacts.** The continual knowledge adaptation mechanism introduced by CKA-RL can substantially improve the data- and compute-efficiency of autonomous systems that must operate in non-stationary environments. By enabling robots, intelligent assistants, and other agents to rapidly integrate new skills without repeatedly retraining from scratch, our method can reduce energy consumption and carbon footprint associated with large-scale model updates.

**Negative Societal Impacts.** As with any advanced continual learning technique, there is a risk that CKA-RL could be exploited to develop adaptive adversarial agents that continuously learn to evade detection or defenses.

