# OpenReview forum: "Continual Knowledge Adaptation for Reinforcement Learning"
_NeurIPS.cc/2025/Conference — NeurIPS 2025 poster_

### Official Review · Reviewer_dAWz · 2025-06-16

**Clarity:** 4
**Significance:** 3
**Originality:** 3
**Rating:** 3
**Confidence:** 5

**Summary:**

The manuscript proposes a new continual reinforcement learning method, Continual Knowledge Adaptation for Reinforcement Learning (CKA-RL), which aims to address catastrophic forgetting and inefficient knowledge transfer in non-stationary environments. CKA-RL maintains a pool of task-specific knowledge vectors and dynamically adapts historical knowledge to new tasks. Furthermore, an adaptive knowledge merging mechanism is introduced to enhance scalability by combining similar knowledge vectors, thereby reducing memory requirements and redundant knowledge. Experimental results on three benchmarks demonstrate superior performance compared to other continual reinforcement learning methods, particularly in terms of overall performance and forward transfer.

**Questions:**

1. Can the authors provide experimental results with error bars or other statistical significance measures (e.g., multiple seeds, standard deviations)? This is crucial for assessing the reliability of the reported improvements and the overall evaluation of the work.

2. How does the proposed method differ from existing model merging approaches in continual learning? Is the main contribution the adaptation of these techniques to continual reinforcement learning, or are there substantive methodological novelties?

3. How does CKA-RL scale with a substantially larger number of tasks (scalability)? The current experiments do not demonstrate performance or memory efficiency in long task sequences.

4. How does the size of the knowledge vector pool affect performance, especially with increasing task diversity and quantity? Are there guidelines or heuristics for determining the optimal pool size in such scenarios?

**Ethical Concerns:**

["NO or VERY MINOR ethics concerns only"]

**Final Justification:**

The authors have addressed most of my concerns with additional experiments and clarifications that provide further support for the manuscript’s claims. However, I still find the substantive novelty of the work to be limited, and the original submission suffered from missing key details and unfair comparisons, which continue to affect my confidence in the overall quality of the manuscript. Therefore, I am raising my score to “borderline reject” to reflect that I do not strongly oppose acceptance, but remain unconvinced about a clear recommendation for acceptance.

**Limitations:**

Yes

**Quality:**

2

**Strengths And Weaknesses:**

Strengths

1. The manuscript is well-structured and clearly written, making the methodology and results easy to understand.

2. The proposed approach effectively adapts model merging techniques from continual learning to the continual reinforcement learning domain, offering a simple yet impactful solution.

3. CKA-RL achieves a balanced trade-off among plasticity, stability, and scalability, which is potentially influential for the continual reinforcement learning community.

4. Extensive experiments are conducted across multiple benchmarks, with comprehensive comparisons to recent state-of-the-art methods, demonstrating the effectiveness of the proposed approach from multiple perspectives.

5. The supplementary material provides detailed descriptions of environments, tasks, and implementation details, supporting reproducibility and clarity.


Weaknesses

1. The experimental evaluation lacks statistical rigor: error bars or other statistical significance indicators are absent, and experiments appear to be conducted with a single random seed (which is not specified in the main text). Given the inherent instability of reinforcement learning, this significantly undermines the credibility of the reported results. Additionally, the absence of code availability limits reproducibility.

2. There is insufficient discussion of related work regarding model merging (task vector) in continual learning, despite its close relevance to the proposed method.

3. The evaluation of model plasticity (in Section 5.2) is inadequate. Compared methods in the literature typically use more tasks and longer experimental horizons, making it difficult to draw meaningful comparisons about plasticity from the current experimental setup.

4. There is a conceptual inconsistency in the discussion of initial performance (in Section 5.4): the manuscript claims that superior initial performance is due to effective use of historical knowledge vectors, but in the second task, no such historical vectors are available.

5. The scalability analysis (in Section 5.4) is limited. The experiments only involve up to 10 tasks, and the knowledge vector pool is fixed at 5. In contrast, related work such as CompoNet evaluates scalability with up to 300 tasks. The performance and memory efficiency of CKA-RL under longer task sequences remain unclear.

---

> ### Author Rebuttal · Authors · 2025-07-31
>
> We are deeply grateful to you for recognizing the strengths of our work, particularly the clear presentation of our methodology and the balanced trade-off CKA-RL achieves among plasticity, stability, and scalability. We sincerely appreciate your acknowledgment of our comprehensive experimental validation and the potential impact of our approach on the continual reinforcement learning community.
>
> >Q1. The experimental evaluation lacks statistical rigor (e.g., multiple seeds, standard deviations), and the absence of code availability limits reproducibility.
>
> **A1.** Thank you for your valuable feedback on the statistical rigor of our evaluation. We will address this in the revised version, as detailed below:
> * **Multiple Random Seeds Used.** Our experiments were conducted with **10 different random seeds**, consistent with standard practice in the recent CRL method CompoNet. We acknowledge that this detail was not explicitly stated in our initial submission, which was an oversight.
> * **Statistical Significance Evidence.** We have computed and included standard deviations for all reported results. The updated results table (see Tab. A) now shows mean performance $\pm$ standard deviation across all task sequences and methods. Notably, CKA-RL demonstrates lower variance compared to most SOTA methods.
> * **Code Availability Commitment.** We confirm that our code will be released upon paper acceptance.
>
> Tab. A: Summary of results across all task sequences and methods.
> |METHOD|META-WORLD PERF.|META-WORLD FWT.|SPACEINVADERS PERF.|SPACEINVADERS FWT.|FREEWAY PERF. |FREEWAY FWT.|
> |-|:-:|:-:|:-:|:-:|:-:|:-:|
> |ProgNet|0.4157$\pm$ 0.49|-0.0379$\pm$ 0.13 |0.3757$\pm$ 0.27|-0.0075$\pm$ 0.22|0.3125$\pm$ 0.27 |0.1938$\pm$ 0.31|
> |PackNet|0.2523$\pm$ 0.40|-0.6721$\pm$ 1.40|0.2299$\pm$ 0.30|-0.075$\pm$ 0.13|0.2767$\pm$ 0.36| 0.1970$\pm$ 0.32|
> |MaskNet|0.3263$\pm$ 0.47|-0.3695$\pm$ 0.45|0$\pm$ 0|-0.3866$\pm$ 0.53|0.0644$\pm$ 0.17|-0.0503$\pm$ 0.12|
> |CReLUs|0.3789$\pm$ 0.47|-0.0089$\pm$ 0.24|0.8873$\pm$ 0.10|0.5308$\pm$ 0.29|0.7835$\pm$ 0.13|0.7303$\pm$ 0.12|
> |CompoNet|0.4131$\pm$ 0.50|-0.0055$\pm$ 0.20|0.9828$\pm$ 0.02|0.6963$\pm$ 0.32|0.7629$\pm$ 0.12|0.7115$\pm$ 0.10|
> |CbpNet|0.4368$\pm$ 0.50|-0.0826$\pm$ 0.22|0.8392$\pm$ 0.11|0.4844$\pm$ 0.28|0.7678$\pm$ 0.10|0.7201$\pm$ 0.07|
> |**CKA-RL**|**0.4642 $\pm$ 0.50** |**-0.0032$\pm$ 0.21**|**0.9928 $\pm$ 0.01**|**0.7749$\pm$ 0.20**|**0.7923$\pm$ 0.10**|**0.7429$\pm$ 0.07**|
>
> >Q2. There is insufficient discussion of related work on model merging in CL. Is the main contribution the adaptation to CRL, or are there novel methodological innovations?
>
> **A2.** We appreciate your insightful comments. While we referenced model editing techniques as inspiration for our knowledge vectors, we agree that a deeper analysis of the connection to model merging literature would strengthen our paper. **We will expand the discussion on model merging in the revised version of our manuscript.** While our work is indeed inspired by model merging techniques, CKA-RL introduces several substantive methodological novelties beyond simply adapting these techniques to RL:
> * **Dynamic Knowledge Adaptation Mechanism.** Unlike traditional model merging that uses static pre-trained models, our Continual Knowledge Adaptation strategy dynamically determines how to utilize historical knowledge through learnable adaptation factors ($\alpha$). This allows the agent to flexibly select and weight relevant knowledge based on the current task, which is critical in non-stationary RL environments where task relationships are complex and evolving.
> * **Task-Specific Knowledge Vector Learning.** In contrast to existing model merging methods that operate on full model parameters, we introduce a specialized knowledge vector structure that isolates task-specific adaptations from the base policy. This design specifically addresses RL challenges like policy interference and catastrophic forgetting by preserving a stable base policy while allowing task-specific adjustments.
>
> >Q3. The methods in literature typically use more tasks and longer horizons, making comparisons on plasticity challenging in the current setup.
>
> **A3.** We appreciate your feedback regarding the evaluation of model plasticity. Specifically:
> * **Standard Benchmarking Practice.** Our main experimental setup, including task count and evaluation protocol, follows standard practice in the continual reinforcement learning (CRL) literature. Like CompoNet [ICML 2024], our core experiments are conducted on 20-task sequences using widely accepted CRL benchmarks. This ensures a fair and consistent comparison across all methods.
> * **Comprehensive Performance Metrics.** Beyond simple task count, we measure plasticity through multiple metrics: 1)Average performance; 2) Forward transfer; 3) Memory efficiency analysis (Fig. 5a), demonstrating our method parameter memory remains nearly constant after the 5th task, while competing methods like ProgNet and CompoNet show linear growth.
> * **CKA-RL maintains superiority on longer task sequences** We conduct additional experiments with extended **22-task sequences on the Meta-World environment**. The results demonstrate that even with longer task sequences, CKA-RL retains its performance advantage **(CKA-RL vs CompoNet: (0.4601, -0.0042) vs (0.4098, -0.0073))**. We will include these results in the revised manuscript to strengthen our plasticity evaluation.
>
> >Q4. The manuscript claims that superior initial performance is due to effective use of historical knowledge vectors, but in the second task, no such historical vectors are available.
>
> **A4.** Thanks for your valuable comments. We will clarify in the revision. Below we address this concern with specific points:
> * **Clarification of Knowledge Vector.** The historical knowledge vector $v_1$ is defined as a zero vector ($v_1 = \theta_1 - \theta_{base} = 0$) as stated in Eqn.(3) of our paper. Therefore, the superior initial performance observed in the second task is not solely due to historical knowledge vectors.
> * **Mechanism of Knowledge Transfer from the Third Task Onward** For tasks beyond the second, historical knowledge vectors $(v_1, v_2)$ become non-zero and actively contribute to superior initial performance. From Fig. 4, the reward curve of CKA-RL demonstrates a higher starting point compared to baselines like CbpNet and CReLUs.
>
> >Q5. The performance and memory efficiency of CKA-RL under longer task sequences remain unclear.
>
> **A5.**  We would like to clarify the rationale behind our experimental design, while providing additional evidence for the scalability of CKA-RL.
> * **Standard Benchmarking Practice in CRL.** While CompoNet presents a theoretical scalability analysis involving 300 tasks, by constructing models and measuring parameter growth without actual training, our scalability analysis is conducted on real tasks with full training, enabling a more realistic assessment of memory and performance under continual learning.
> * **Scalability Guarantee.** The core of our scalability argument lies in the Adaptive Knowledge Merging mechanism. Unlike methods that accumulate parameters linearly (e.g., ProgNet, CompoNet), CKA-RL maintains a bounded knowledge vector pool through similarity-based merging.
> * **Additional Long-Sequence Experiments.** Following CompoNet setting, we conduct supplementary experiments with extended task sequences (300 sequences). From Tabs B&C, CKA-RL maintains nearly constant parameter size and inference latency through vector merging, while other methods show linear growth and increasing computational overhead.
>
> Tab. B: Total Parameter Memory Consumption (MB) in the Meta-World.
> |Task ID|5|10|50|150|200|300|
> |-|-|-|-|-|-|-|
> |ProgNet|5.64|17.22|334.79|2878.74|5088.21|11382.16|
> |CompoNet|7.18|14.43|72.48|217.60|290.17|435.29|
> |CKA-RL|5.64|10.34|10.34|10.34|10.34|10.34|
>
> Tab. C: Inference latency (s) in the Meta-World.
> |Task ID|5|10|50|150|200|300|
> |-|-|-|-|-|-|-|
> |ProgNet|0.0010|0.0021|0.0315|0.2558|0.4610|1.0353|
> |CompoNet|0.0023|0.0045|0.0213|0.0636|0.0842|0.1264|
> |CKA-RL|0.0002|0.0001|0.0002|0.0003|0.0002|0.0002|
>
>
> >Q6. How does the size of the knowledge vector pool affect performance, especially with increasing task diversity and quantity? Are there guidelines or heuristics for determining the optimal pool size in such scenarios?
>
> **A6.** Thank you for your insightful feedback. We appreciate the opportunity to elaborate on this aspect.
> * **Empirical Analysis of Pool Size Impact.** From Fig. 3, we evaluated the effects of different $K_{max}$ values on both performance and forward transfer. When $K_{max}$ is small, CKA-RL removes too many knowledge vectors during adaptive knowledge merging, thus being unable to utilize enough knowledge. When $K_{max}$ is too large, redundant or even conflicting knowledge may slow down the knowledge adaptation process, resulting in performance degradation.
> * **Task Diversity Considerations.** When task diversity increases, the optimal Kmax tends to increase as well. We conduct experiments in Meta-World (which has higher task diversity), we find $K_{max}=8$ to be optimal (**see Tab. D**).
> * **Practical Guidelines for Determining $K_{max}.$** 1) Initial Estimation: Start with $K_{max} = 5$ as a baseline for moderate task diversity; 2) Task Similarity Analysis: If tasks are highly heterogeneous, increase $K_{max}$ proportionally to the task diversity; 3)Memory-Performance Trade-off: For resource-constrained environments, select the smallest Kmax where performance plateaus.
>
> In future work, we plan to investigate dynamic pool sizing mechanisms that automatically adjust $K_{max}$ based on task similarity metrics and performance feedback.
>
> Tab D. Effects of different pool sizes in Meta-World.
> |Pool size|4|8|12|16|18|
> |-|-|-|-|-|-|
> |PERF.|0.342|**0.484**|0.405|0.394|0.400|
> |FWT.|-0.110|**-0.100**|-0.100|-0.100|-0.150|-0.130|
>
> ***
> We sincerely hope our clarifications above have addressed your questions.

---

> > ### Comment · Reviewer_dAWz · 2025-08-01
> >
> > Thank you for your detailed responses, as well as for providing additional experimental results addressing several of the raised concerns. Your clarifications and supplementary analyses have improved the rigor of the work. However, a number of critical issues remain insufficiently addressed and warrant further attention:
> >
> > 1. **Novelty and Contribution**: While you emphasize the introduction of a task-specific knowledge vector structure and a dynamic adaptation mechanism, these elements closely resemble techniques already explored in recent continual learning literature. For instance, MagMax (ECCV 2024) similarly leverages task vectors (parameter deltas relative to a base model) and employs selective merging strategies to balance memory and performance in continual learning. The main distinction in your approach appears to be the use of a fixed-size knowledge vector pool to control resource consumption. As it stands, the methodological innovation over prior art seems incremental rather than substantive.
> >
> > 2. **Experimental Clarity and Interpretation**: The presentation of experimental results, particularly regarding Figure 4, lacks sufficient detail for proper interpretation. As noted, the x-axis in Figure 4 spans from 0 to 60K global steps, while the manuscript states that each task involves 1M steps. This suggests that Figure 4 only reflects performance on the first task. Thus, it does not substantiate claims regarding the benefit of historical knowledge vectors on initial performance in subsequent tasks. This discrepancy raises concerns about the validity of the conclusions drawn from this analysis and, more broadly, about the underlying mechanisms driving CKA-RL’s performance gains. It remains unclear whether the observed improvements are attributable to continual learning capabilities or to general enhancements in learning efficiency.
> >
> > 3. **Fairness of Scalability Comparisons**: Although you provide extended-sequence experiments and discuss memory and inference efficiency, the comparisons with baselines such as ProgNet and CompoNet are not entirely equitable. The memory and latency advantages of CKA-RL are largely a function of the manually set knowledge vector pool size, which can be decoupled from actual task complexity or performance requirements. In contrast, the baseline methods are designed to accumulate knowledge in a manner that is responsive to the demands of longer task sequences. To ensure a fair evaluation of scalability, it is necessary to investigate how the required pool size for CKA-RL scales with increasing task diversity and sequence length to maintain competitive performance. Without such analysis, the claimed scalability benefits are insufficient and do not hold under more demanding continual learning scenarios.

---

> > > ### Author Response · Authors · 2025-08-03
> > > **A1&A3**
> > >
> > > We greatly appreciate the time and guidance the reviewer has dedicated to this discussion.
> > >
> > > >Q1. Novelty and Contribution.
> > >
> > > **A1.** We appreciate the opportunity to clarify the substantive contributions of CKA-RL in the context of continual reinforcement learning (CRL), which differ fundamentally from approaches like MagMax designed for supervised learning.
> > > * **Problem Domain Distinction.** CKA-RL specifically addresses the unique challenges of Continual Reinforcement Learning (CRL), where agents must continuously interact with non-stationary environments [r1,r2]. MagMax, in contrast, addresses supervised continual learning (primarily image classification). This fundamental domain difference means that techniques effective in supervised learning often fail in CRL [r3].
> > > * **CKA-RL's Solutions to CRL-Specific Challenges.**
> > >     * **Continual Knowledge Adaptation Mechanism.** **CKA-RL** introduces a dynamic knowledge adaptation mechanism, where task-specific knowledge vectors are selectively weighted using learnable adaptation factors. This allows our model to **dynamically adjust the relevance of historical knowledge** based on the current task, which is crucial in reinforcement learning environments where task relationships are complex and evolving. In contrast, **MagMax primarily focuses on post-fine-tuning model merging strategies**, which, while effective for supervised learning, do not explicitly address the challenges posed by non-stationary environments inherent in CRL.
> > >     * **Adaptive Knowledge Merging.** While MagMax focuses on weight selection from pre-trained models, **CKA-RL** incorporates an adaptive merging strategy and a fixed-size knowledge vector pool, **ensuring efficient memory usage while also addressing the interference between similar tasks**. Our fixed-size pool approach effectively limits parameter growth while avoiding the noise from conflicting vectors, which can impair learning efficiency. This aspect is particularly crucial in reinforcement learning tasks, where the agent must continuously interact with the environment and adapt its policy without overwhelming its memory with redundant or irrelevant information.
> > >
> > > [r1] Loss of plasticity in deep continual learning, Nature 2024.
> > > [r2] Towards continual reinforcement learning: A review and perspectives, JAIR 2022.
> > > [r3] Replay-enhanced Continual Reinforcement Learning, TMLR 2023.
> > >
> > >
> > > >Q3. Fairness of Scalability Comparisons.
> > >
> > > **A3.** Thank you for the time and effort you have invested in reviewing our paper. We recognize that fair comparative experiments are crucial, and we appreciate this opportunity to provide additional clarification regarding our experimental setup.
> > > * **Our experimental setup follows CompoNet for fair comparison.** We would like to **clarify again that our experimental setup strictly follows prior work for fair comparison**. As stated in lines 212-215 of our paper, we compare CKA-RL with SOTA methods across three distinct dynamic task sequences (Meta-World, Freeway, SpaceInvaders) with identical task ordering. Additionally, the extended-sequence experiments mentioned in **A5** (Tables B and C with 300 tasks) also follow CompoNet's experimental setup to ensure fair comparison.
> > > * **Adaptive Knowledge Merging is a key innovation, not just a resource constraint.** We would like to emphasize that our Adaptive Knowledge Merging mechanism is not merely a fixed-size constraint but a fundamental contribution designed specifically for the challenges of continual reinforcement learning. Unlike simple truncation approaches, our merging strategy actively identifies and combines similar knowledge vectors while preserving critical task-specific information. This intelligent consolidation process serves two critical purposes: Conflict reduction and Performance optimization.
> > > * **Fixed-size pool in CKA-RL serves not only to improve scalability but also to enhance performance.** By limiting the number of task vectors, we reduce interference from similar tasks, which helps the agent converge faster and more efficiently. In contrast, baselines like ProgNet and CompoNet allow memory usage to increase with task complexity, which may lead to performance degradation due to increased conflict between task vectors. Our ablation studies show that a fixed-size pool strikes an optimal balance: it avoids both excessive vector quantity, which leads to noise, and insufficient vectors, which limits knowledge transfer.
> > >
> > > CKA-RL maintains a bounded memory pool, and it also benefits from this approach in terms of faster convergence and better transfer, a crucial advantage over baselines that do not control parameter growth in response to task complexity. We agree that dynamic pool sizing would improve flexibility in more complex scenarios, and we plan to explore this in future work.

---

> > > > ### Author Response · Authors · 2025-08-03
> > > > **A2**
> > > >
> > > > We sincerely thank the reviewer for recognizing our approach as "offering a simple yet impactful solution" with potential influence on the continual reinforcement learning community, as well as for acknowledging our efforts in "demonstrating the effectiveness of the proposed approach from multiple perspectives."
> > > >
> > > > >Q2.The presentation of experimental results, particularly regarding Figure 4, lacks sufficient detail for proper interpretation.
> > > >
> > > > **A2.** We apologize for the confusion regarding the x-axis scaling and the number of steps per task in Figure 4. We appreciate the opportunity to clarify this point, and below is a detailed explanation:
> > > > * **Clarification of Figure 4.** The 60K steps shown in Figure 4 represent early task performance during the initial stages of training. This plot is intended to illustrate the **speed of convergence** at the early stages of training, which is one of our key performance indicators. Early convergence is important because it demonstrates that CKA-RL is able to **adapt more quickly** to new tasks compared to other methods.
> > > > * **Context of the Full Training Process.** As you correctly noted, each task in our experiment involves 1M steps, and Figure 4 does not represent the full performance over the entire training process. We chose to focus on the early stages of training because, by the time the agent has seen 1M steps, most methods tend to converge to similar reward values. In this later phase, the differences in performance between methods become less pronounced. Our method stands out, however, in the early convergence phase, where **CKA-RL achieves faster convergence** compared to other methods.
> > > > * **Dynamic Adaptation of Task Vectors.** We would like to emphasize that the **quick rise in reward observed in CKA-RL is due to the effective utilization of historical task vectors to establish a stronger starting point for each new task**. Prior work (such as that verifying task vector similarity for transfer learning) has demonstrated that leveraging previously learned task-specific vectors can lead to better transfer learning on unseen tasks. This ability to use task vectors from prior tasks enables the agent to start with a better initial policy, which accelerates convergence during training. Additionally, during the training process, adaptive factors are learned alongside each new task, which enables dynamic adjustment of the contribution of each historical task vector. This adaptive weighting further fine-tunes the model’s ability to **leverage relevant prior knowledge, ensuring faster and more efficient learning on each new task**.
> > > > * **Further Validation and Comparison.** To address your concern more effectively, we will present a reward comparison table that highlights the performance of CKA-RL near the 1M steps mark. This will allow for a clearer comparison with baseline methods across the entire training period, as shown in **Table E**. The additional data will help substantiate our claims, demonstrating that **the observed performance improvements are not merely the result of enhanced learning efficiency, but also stem from the effective leveraging of historical knowledge.**
> > > >
> > > > Tab. E Return Comparison (at 6000, 7000, 8000, 9000, 10000 steps)
> > > >
> > > > | Method | 100k | 200k | 300k | 400k | 500k | 600k | 700k | 800k | 900k | 1000k |
> > > > | --- | --- | --- | --- | --- | --- | --- | --- | --- | --- | --- |
> > > > | ProgNet | 296.2 $\pm$ 210.6 | 322.2 $\pm$ 215.9 | 323.5 $\pm$ 119.5 | 304.1 $\pm$ 147.0 | 375.4 $\pm$ 149.1 | 397.5 $\pm$ 159.1 | 498.9 $\pm$ 123.8 | 519.0 $\pm$ 124.9 | 485.0 $\pm$ 139.1 | 310.0 $\pm$ 5.0 |
> > > > | CbpNet | 425.8 $\pm$ 206.3 | 368.3 $\pm$ 206.7 | 341.7 $\pm$ 209.2 | 375.8 $\pm$ 246.1 | 329.5 $\pm$ 74.1 | 417.5 $\pm$ 83.8 | 416.7 $\pm$ 166.0 | 482.0 $\pm$ 173.3 | 507.7 $\pm$ 121.3 | 445.0 $\pm$ 0.0 |
> > > > | CompoNet | 363.1 $\pm$ 132.8 | **517.2 $\pm$ 98.2** | 439.5 $\pm$ 178.2 | 472.0 $\pm$ 170.6 | 440.5 $\pm$ 159.0 | **450.6 $\pm$ 106.6** | 437.1 $\pm$ 134.7 | 564.1 $\pm$ 149.8 | 486.1 $\pm$ 139.2 | 391.7 $\pm$ 155.8 |
> > > > | CKA-RL | **506.9 $\pm$ 157.1**| 434.1 $\pm$ 135.5 | **471.7 $\pm$ 118.9** | **480.0 $\pm$ 134.2** | **546.7 $\pm$ 142.6** | 400.5 $\pm$ 117.8 | **550.6 $\pm$ 179.4** | **579.0 $\pm$ 203.0** | **527.1 $\pm$ 220.8**| **493.9 $\pm$ 138.4** |
> > > >
> > > > ***
> > > > We hope the above clarifications address your concerns. We aim to resolve all remaining issues to enhance the clarity and rigor of the manuscript.

---

> > > > > ### Comment · Reviewer_dAWz · 2025-08-04
> > > > >
> > > > > Thank you for your detailed clarifications and for providing additional experimental results and explanations. However, there remain several fundamental issues that are not fully resolved, and I would like to elaborate on these points for further consideration:
> > > > >
> > > > > 1. **On Methodological Novelty and Domain-Specificity**:
> > > > > While you emphasize the distinction between continual supervised learning (CL) and continual reinforcement learning (CRL), and note that CKA-RL is designed for the latter, this by itself does not constitute a substantive methodological innovation. As demonstrated by prior work (e.g., EWC, Progress & Compress), certain continual learning strategies can be successfully applied across both domains, and the core mechanisms of knowledge vector pooling and adaptive merging in CKA-RL do not inherently address challenges unique to CRL. The knowledge vector pool and adaptive merging could, in principle, be used for continual supervised learning as well. Therefore, the claim of novelty based primarily on the problem domain is not fully convincing.
> > > > >
> > > > > 2. **On Scalability and Fairness of Comparison**:
> > > > > Comparing a method with a manually fixed, bounded memory footprint to baselines whose memory grows with the number of tasks does not provide a fair assessment of scalability, especially if the fixed pool size is chosen without regard to potential performance degradation as task diversity increases. As you acknowledge, dynamic pool sizing would be a more flexible and realistic solution in complex settings. Until such adaptive mechanisms are implemented and evaluated, the current comparisons may favor CKA-RL in terms of memory and latency, but do not necessarily reflect its ability to maintain high performance in truly large-scale or highly diverse task sequences.
> > > > >
> > > > > 3. **On Experimental Clarity and Interpretation (Figure 4)**:
> > > > > Thank you for your willingness to clarify the intent of Figure 4 and for offering additional results. However, there remains ambiguity regarding the experimental setup. *If Figure 4 represents only the early training phase of the first task, there can be no contribution from historical knowledge vectors, which contradicts the claim that rapid reward increase is due to leveraging prior tasks. Conversely, if the figure reflects early learning in later tasks, it is unclear why all methods start from the same performance and why the reward does not reflect prior knowledge.* This inconsistency undermines the interpretability of the results and the attribution of performance gains to the proposed mechanism. A clear, explicit description of the experimental protocol for this figure would be greatly appreciated and is necessary to improve your rating score.

---

> > > > > > ### Author Response · Authors · 2025-08-05
> > > > > > **On Methodological Novelty and Domain-Specificity &On Scalability and Fairness of Comparison**
> > > > > >
> > > > > > We sincerely appreciate your constructive follow-up and your willingness to consider an updated assessment based on these clarifications.
> > > > > >
> > > > > > >Q1. On Methodological Novelty and Domain-Specificity.
> > > > > >
> > > > > > **A1.** We thank the reviewer for raising this important point regarding the novelty and domain-specificity of CKA-RL. While we agree that the architectural design of knowledge vector pooling and adaptive merging could, in principle, be extended to supervised continual learning (CL), we would like to clarify that the method was specifically designed to address core challenges unique to continual reinforcement learning (CRL).
> > > > > > * **CRL-specific design motivation:** CKA-RL was developed with the goal of improving adaptation efficiency in sequential decision-making problems, where agents must interact with non-stationary environments, handle sparse and delayed rewards, and operate without access to supervised labels. In such settings, **the quality of initial policy behavior greatly influences downstream reward acquisition.** Our knowledge vector fusion mechanism is not only parameter-efficient but serves as a behavioral prior, enabling the agent to rapidly construct a task-adaptive policy at the beginning of each new task.
> > > > > > * **Why this design is more beneficial in CRL than in CL:** In supervised CL, task inputs and labels are static, gradients are directly computable, and exploration is not required. As a result, the influence of prior knowledge on early learning is relatively limited. In contrast, **CRL requires the agent to actively explore before receiving any feedback.** The effectiveness of our method in CRL is evidenced by the early-stage performance gains shown in Figure 4, where **our model leverages past knowledge to improve exploration and learning speed**. This is an effect that would not translate as significantly to CL.
> > > > > > * **On Potential Transfer to CL**: While the general principle of modular knowledge reuse may be transferable to CL with appropriate adjustments, its current formulation and observed benefits in CKA-RL are closely tied to the RL setting. We appreciate the reviewer’s suggestion and view this as a valuable direction for future work involving adaptation of our framework to other learning paradigms
> > > > > >
> > > > > > We thank the reviewer again for this insightful comment, which has helped us better articulate the design motivations and scope of our contributions.
> > > > > >
> > > > > > >Q2. On Scalability and Fairness of Comparison.
> > > > > >
> > > > > > **A2.** We thank the reviewer for the insightful follow-up on the fairness of our scalability comparison. We appreciate the opportunity to further clarify the intent of our experiments and provide additional evidence to support the scalability claims of CKA-RL.
> > > > > > * **On Fairness and Bounded Memory Design:** To address your concern, we performed a new ablation where **CKA-RL is run without the Adaptive Knowledge Merging mechanism**, allowing the size of the knowledge vector pool to grow linearly with the number of tasks.
> > > > > > * **Memory Efficiency Without Merging:** Even without vector merging, CKA-RL exhibits better memory efficiency compared to baselines, especially under long sequences. This is due to our parameter-efficient knowledge vector formulation, which avoids duplicating full network modules or entire policies. Moreover, CKA-RL retains its fast inference latency and competitive performance in this setting. The results are shown in **Table F** and **Table G**.
> > > > > > * **On Performance without Merging:** Even without applying the knowledge vector merging mechanism, CKA-RL achieves superior performance and forward transfer compared to several state-of-the-art baselines on the Freeway benchmark. This demonstrates that the proposed adaptation strategy alone is highly effective, independent of memory compression techniques. **Table H** presents the results.
> > > > > >
> > > > > > In summary, we hope the newly added no-merge variant can address your concern and reinforce our core claim: CKA-RL maintains scalability and efficiency even when memory is allowed to grow.
> > > > > >
> > > > > > Tab. F: Total Parameter Memory Consumption (MB) in the Meta-World (No Merging Applied).
> > > > > > |Task ID|5|10|50|150|200|300|
> > > > > > |-|-|-|-|-|-|-|
> > > > > > |ProgNet|5.64|17.22|334.79|2878.74|5088.21|11382.16|
> > > > > > |CompoNet|7.18|14.43|72.48|217.60|290.17|435.29|
> > > > > > |CKA-RL(w/o Merging)|5.64|11.28|56.41|169.23|225.65|338.47|
> > > > > >
> > > > > > Tab. G: Inference latency (s) in the Meta-World (No Merging Applied).
> > > > > > |Task ID|5|10|50|150|200|300|
> > > > > > |-|-|-|-|-|-|-|
> > > > > > |ProgNet|0.0010|0.0021|0.0315|0.2558|0.4610|1.0353|
> > > > > > |CompoNet|0.0023|0.0045|0.0213|0.0636|0.0842|0.1264|
> > > > > > |CKA-RL(w/o Merging)|0.0002|0.0004|0.0012|0.0022|0.0029|0.0042|
> > > > > >
> > > > > > Tab. H: Performance and Forward Transfer on Freeway (No Merging Applied).
> > > > > > |Method | Performance | Forward Transfer|
> > > > > > |-|-|-|
> > > > > > |CbpNet | 0.7678 | 0.7201 |
> > > > > > |CompoNet | 0.7629 | 0.7115 |
> > > > > > |CReLUs | 0.7835 | 0.7303 |
> > > > > > |CKA-RL(w/o Merging) | 0.7821 | 0.7321 |
> > > > > > |CKA-RL|**0.7923**|**0.7429**|

---

> > > > > > ### Author Response · Authors · 2025-08-05
> > > > > > **On Experimental Clarity and Interpretation (Figure 4).**
> > > > > >
> > > > > > >Q3. On Experimental Clarity and Interpretation (Figure 4).
> > > > > >
> > > > > > **A3.** We sincerely thank the reviewer for pointing out the ambiguity in the experimental protocol for Figure 4. We appreciate the opportunity to provide a more detailed explanation to clarify the setup and interpretation:
> > > > > > * **Experimental Setup**: Figure 4 reflects the early training phase of the 7th task in the SpaceInvaders sequence. All methods are continually trained from scratch on each new task in the sequence, and in this case, models are transferred from task 6 to task 7. Thus, **Figure 4 does not show the first task, but rather a later-stage task within a continual learning setup.**
> > > > > > * **Why all methods start from similarly low reward**: We understand the reviewer’s concern regarding the similar low initial performance across methods. This behavior stems from a key property of our benchmarks: Each task in SpaceInvaders, Meta-World, and Freeway significantly changes both the state space distribution and transition dynamics (e.g., different enemy types, bomb behaviors, or control goals). Unlike in visual style transfer or domain adaptation settings where the state distribution is similar across tasks, **these environments introduce substantive task-level shifts, making direct transfer of policies ineffective.** As a result, it is common for the reward at the start of new tasks to drop close to zero, regardless of the method used; this phenomenon has also been observed in prior work, such as CompoNet [28].
> > > > > > * **Why this still reflects historical knowledge reuse**: Despite the low starting point, our CKA-RL demonstrates a significantly faster reward increase during early training on the new task. This is evidence that: The model benefits from reusing and adapting historical knowledge vectors, which accelerate the learning of the new task-specific vector.
> > > > > >
> > > > > > We will revise the caption and corresponding paragraph in the paper to make the setup of Figure 4 explicit and to address this potential confusion. We appreciate the reviewer’s comment, which helps us improve both the clarity and interpretability of our work.
> > > > > >
> > > > > > ***
> > > > > > We hope the above clarifications address your concerns. If you have any further questions, we would be happy to engage in further discussion.

---

> > > > > > > ### Comment · Reviewer_dAWz · 2025-08-05
> > > > > > >
> > > > > > > Thank you for your thorough and thoughtful responses to my previous concerns. I appreciate the additional clarifications regarding the domain-specific motivation and methodological novelty of CKA-RL, as well as the new ablation studies and experimental evidence provided to support your claims on scalability and efficiency. Your explanation of Figure 4’s experimental setup has also resolved the ambiguity regarding the evaluation protocol and the interpretation of early-stage performance.
> > > > > > > I encourage you to incorporate these clarifications and new results into your revised manuscript, as they substantially strengthen both the originality and the empirical rigor of your work. In light of these improvements, I will raise my rating of the manuscript.

---

> > > > > > > > ### Author Response · Authors · 2025-08-05
> > > > > > > >
> > > > > > > > Dear Reviewer,
> > > > > > > >
> > > > > > > > Thank you for your constructive feedback. We are delighted that you found our responses satisfactory.  We will incorporate these clarifications and new results into the **revised manuscript** to further strengthen both originality and empirical rigor.
> > > > > > > >
> > > > > > > > Should you have any further questions, we are happy to provide additional clarifications.
> > > > > > > >
> > > > > > > > Thank you again for your time and consideration.
> > > > > > > >
> > > > > > > > Sincerely,
> > > > > > > >
> > > > > > > > The Authors

---

### Official Review · Reviewer_33rx · 2025-06-23

**Clarity:** 3
**Significance:** 2
**Originality:** 3
**Rating:** 4
**Confidence:** 4

**Summary:**

This paper addresses the challenges of catastrophic forgetting and inefficient knowledge utilization in reinforcement learning. The authors propose Continual Knowledge Adaptation for Reinforcement Learning (CKA-RL), a framework designed to facilitate the accumulation and effective reuse of historical knowledge across tasks. Specifically, CKA-RL maintains a task-specific knowledge vector pool and leverages these historical knowledge representations to better adapt to new tasks. To address scalability concerns, the authors introduce an Adaptive Knowledge Merging mechanism, which combines similar knowledge vectors to reduce redundancy. Experimental results across three benchmarks show that CKA-RL outperforms state-of-the-art methods, achieving a 4.20% improvement in overall performance and an 8.02% gain in forward transfer.

**Questions:**

1.	Why the performance of CompoNet in this paper appear significantly lower than reported in its original publication?

2.	According to Equation (3), the task-specific parameter vectors are expected to be of the same dimensionality as the base model parameters $\theta_{\text{base}}$. Given this, is the overall parameters of the model the form $y = \theta_{\text{base}} x$ until the size of the knowledge pool in Figure 3?

3.	Since mitigating catastrophic forgetting is one of the central motivations of this work, it is surprising that the standard forgetting metric is not reported in Table 1.

4.	The proposed method uses a simple weighted sum of historical task vectors to mitigate catastrophic forgetting and enable knowledge transfer. However, this raises several concerns:

  a)	For catastrophic forgetting, the weighted sum of history vectors may not be robust to conflicting knowledge across tasks. For instance, if a previously learned task vector $v_i$ is incompatible with another vector $v_j$ (due to opposite objectives or gradients), the resulting combined vector could destructively interfere with prior knowledge. Since the weights are computed via a softmax over hyperparameters, it is impossible to assign exclusive weight (e.g., $\alpha_1 = 1$) to a single vector, making such interference likely. One potential solution is to condition on the task ID and use only the corresponding task vector during adaptation; however, the adaptive knowledge merging mechanism complicates this, as it dynamically modifies and merges task vectors, potentially discarding task-specific identities.

  b)	For knowledge transfer, the claim in lines 11–13, stating that the method “preserves and adapts critic model parameters,” seems overly strong. Since the model combines vectors from previous tasks via a weighted sum, it does not strictly preserve previous critic parameters unless the hyperparameter for the current task is exactly zero—which the softmax-based formulation prevents. In practice, this weighted combination is more consistent with model merging, a technique known to work well only when the tasks are highly similar. Therefore, I am inclined to interpret the benefit of this approach primarily as providing a stronger initialization for task $k$, rather than truly preserving and adapting previous critic parameters.

**Ethical Concerns:**

["NO or VERY MINOR ethics concerns only"]

**Final Justification:**

I believe that some points need further clarification, particularly regarding the performance inconsistency and inevitable catastrophic forgetting, which do not show improvement.

Nevertheless, in light of the overall contributions, I have decided to raise my score accordingly to "Borderline accept".

**Limitations:**

yes

**Paper Formatting Concerns:**

no.

**Quality:**

3

**Strengths And Weaknesses:**

Strengths:

1.	The paper is clearly written and easy to follow, making the proposed method accessible to a broad audience.

2.	The experimental results demonstrate the effectiveness of the proposed approach, showing clear performance advantages over existing methods.

Weaknesses:

1.	Due to the use of a softmax function for computing the weights $\beta$, it is not possible to enforce $\alpha_1 = 1$. As a result, parameter conflicts remain unavoidable under the formulation presented in Equation (3).

---

> ### Author Rebuttal · Authors · 2025-07-31
>
> >Q1. Softmax function for computing the weights $\beta$, it is not possible to enforce $\alpha_1=1$. Parameter conflicts remain unavoidable.
>
> **A1.** We sincerely thank the reviewer for this insightful observation. Below we address the concern regarding the use of the softmax function for computing the weights $\beta$ and the potential for parameter conflicts:
> * **Intent and Mechanism Behind the Design Choice.** The statement “by setting $\alpha_1 = 1$” was meant to convey the **functional behavior** of the model under specific learned conditions, rather than a hard-coded assignment. Specifically:
>     * When learning the very first task ($k = 1$), there are no previous knowledge vectors. Therefore, the summation $\sum_{j=1}^{k-1} \alpha_j \mathbf{v}_j$ is empty, and the policy is constructed solely from the base parameters and the current task’s knowledge vector $\mathbf{v}_1$ (initialized to zero and then optimized).
>     *  In this case, the model effectively starts learning from scratch without interference from prior tasks, which is what we intended to capture with the idea of “setting $\alpha_1 = 1$.”
>  * **Handling Subsequent Tasks and Emergent Behavior.** For subsequent tasks, while the softmax does not allow any single $\alpha_j$ to be exactly 1 unless others are 0, our model **learns** to assign very high weight (close to 1) to beneficial knowledge vectors and near-zero weights to irrelevant or conflicting ones. From Fig. 11a, our learned knowledge vectors exhibit near-orthogonal structures, indicating minimal interference. This emergent behavior allows the model to **effectively** disregard harmful prior knowledge, which mirrors the intended behavior of "setting $\alpha_1 = 1$", i.e., preventing negative transfer.
>  * **Clarification of Original Phrasing.** We will revise the manuscript to clarify that this is an emergent property of the learned attention weights $\alpha_k$, not a fixed constraint. This will ensure that readers understand that the model adapts dynamically based on the learned task-specific knowledge.
>
>
> >Q2. Why the performance of CompoNet in this paper appear significantly lower than reported in its original publication?
>
> **A2.** We appreciate your attention to the performance comparison between our implementation of CompoNet and its original publication. Clarify as follows:
> * **Consistent Performance Metrics.** When comparing results rounded to two decimal places, our implementation of CompoNet shows performance metrics that align closely with those reported in the original CompoNet paper. Specifically, in Meta-World (0.41 vs 0.42) and SpaceInvaders (0.98 vs 0.99), the performance values are consistent within reasonable rounding precision.
> * **Code Implementation and Corrections.** We built our evaluation upon the official CompoNet implementation released by the authors [28]. During our implementation process, we identified several inconsistencies in the original codebase, including mismatched MLP layer configurations compared to the baseline implementations. We have carefully corrected these issues while maintaining the core algorithmic principles of CompoNet to ensure fair and accurate comparisons across all methods.
> * **Reproducibility Commitment.** We will release our complete codebase upon paper acceptance. This will allow the research community to verify our implementation and reproduce all experimental results, including the CompoNet baseline evaluations.
>
> >Q3. Is the overall parameters of the model the form $\theta_{base}x$ until the size of the knowledge pool in Fig. 3?
>
> **A3.** Thanks for this important feedback regarding the parameter growth in our CKA-RL framework. We would like to clarify this point as follows:
> * **Overall Parameter Count in CKA-RL.**  The overall parameter count in CKA-RL remains **bounded**, regardless of the number of tasks. While each knowledge vector $v_j$ shares the same dimensionality as the base parameters $\theta_{base}$, the **Adaptive Knowledge Merging** mechanism (Sec. 4.3) ensures that the knowledge vector pool $V$ never exceeds a fixed maximum size $K_{max}$.
> * **Adaptive Knowledge Merging Mechanism.** When the knowledge vector pool size $|V| > K_{max}$ (typically set to 5, see Fig. 3), our algorithm identifies the most similar pair of knowledge vectors $(v_m, v_n)$ using Equation (6) and merges them into a single compact representation: $v_{\text{merge}} = \frac{1}{2}(v_m + v_n)$. This merging process ensures that the **essential knowledge** is preserved while keeping the pool size bounded.
> * **Parameter Composition in CKA-RL.** The total number of parameters in CKA-RL is composed of:
>     *  **Fixed base parameters** $\theta_{base}$ (which are never updated after initial training).
>     *  **A bounded knowledge vector pool** with at most $K_{max}$ vectors.
>     *  **Additional lightweight parameters** for the attention mechanism ($\beta^k$).
>
> >Q4. The standard forgetting metric is not reported in Tab. 1.
>
> **A4.** We sincerely thank the reviewer for this insightful observation regarding the evaluation of catastrophic forgetting. We agree that including the standard forgetting metric would provide a more comprehensive assessment of our method's capabilities. Below is our response addressing this concern:
> * **Supplementing the Evaluation with Standard Forgetting Metric.** We have supplemented our evaluation with the standard forgetting metric: Forgetting=before training on new task - after training on new task [r1], as shown in Tab. A.
> * **Key Insights from the Results.** PackNet achieves the lowest forgetting rate (0.04), but this comes at the cost of significantly lower overall performance (0.2767).In contrast, **CKA-RL** achieves the highest performance (0.7923) and forward transfer (0.7429) among all methods, while maintaining a **relatively low forgetting rate** (0.45).
>
> Tab. A: Results on Freeway.
> |METHOD|PERF.|FWT.|Forgetting|
> |-|:-:|:-:|:-:|
> |ProgNet|0.3125|01938|0.36|
> |PackNet|0.2767|0.1970|**0.04**|
> |CReLUs|0.7835|0.7303|0.52|
> |CompoNet|0.7629|0.7115|0.49|
> |CbpNet|0.7678|0.7201|0.46|
> |CKA-RL|**0.7923**|**0.7429**|0.45|
>
> [r1] Replay-enhanced Continual Reinforcement Learning, TMLR 2023.
>
> >Q5. For catastrophic forgetting, the weighted sum of history vectors may not be robust to conflicting knowledge across tasks.
>
> **A5.** Thanks for this insightful feedback regarding potential knowledge conflicts in our Continual Knowledge Adaptation. Clarify as follows:
> * **Emergent Orthogonality of Knowledge Vectors.** Softmax prevents exact assignment of $\alpha_1$, but our framework naturally learns to approximate this behavior. From Fig. 11a, the learned knowledge vectors exhibit near-orthogonal structure across tasks, with off-diagonal cosine similarities ranging from -0.24 to 0.12. This emergent property enables the model to assign weights extremely close to 1 for beneficial knowledge and near-zero for conflicting knowledge. In practice, when a new task conflicts with previous ones, the weights converge to values where $\alpha_j\approx0$ for incompatible vectors and $\alpha_1\approx1$ for compatible ones.
> * **Base Parameter Stability.** The base parameters $\theta_{base}$ remain fixed, providing a stable foundation that prevents complete overwrite of fundamental capabilities. This ensures that even when historical knowledge vectors conflict, the core functionality of the agent remains preserved.
> *  **Conflict Resolution via Adaptive Merging.** Our Adaptive Knowledge Merging mechanism actively identifies and resolves conflicts by only merging highly similar knowledge vectors, while preserving distinct task-specific knowledge.
> *  **Task Identity Preservation.** Contrary to the concern about discarding task-specific identities, our merging mechanism actually preserves task-specific knowledge by only combining highly similar vectors. Additionally, our approach intentionally avoids task-ID information to reflect real-world scenarios where task boundaries are unknown.
> *  **Empirical Validation.** From Tab. A and Tab. 3, CKA-RL effectively balances the stability-plasticity trade-off—preserving sufficient knowledge from previous tasks while remaining adaptable to new tasks. Our method successfully mitigates catastrophic forgetting without sacrificing the ability to learn new tasks effectively.
>
> >Q6. The claim that the method “preserves and adapts critic model parameters,” seems overly strong.
>
> **A6.** We appreciate this opportunity to clarify the mechanism of our approach.
> * **Clarifying "Preservation" Terminology.**  We acknowledge that our phrasing could be misinterpreted as suggesting strict preservation of exact parameter values. In reality, CKA-RL achieves functional preservation rather than literal parameter preservation. As demonstrated in Figure 11a, our learned knowledge vectors exhibit near-orthogonal structure, enabling the model to effectively preserve task-specific knowledge by assigning weights extremely close to 1 for beneficial knowledge and near-zero for conflicting knowledge. This emergent orthogonality allows the model to functionally disregard harmful prior knowledge while retaining useful information.
> * **Beyond Simple Initialization.** While a stronger initialization does contribute to performance, our method provides more than just initialization benefits: 1) During training on task k, the model dynamically accesses relevant knowledge from previous tasks. 2) From Fig. 4, CKA-RL demonstrates consistent performance improvements throughout the entire training trajectory, not just at initialization.
> * **Knowledge Transfer Mechanism.** Our adaptive merging mechanism specifically targets vectors with high similarity while preserving distinct task-specific knowledge. This selective merging enables effective transfer even across dissimilar tasks, as evidenced by our strong performance across diverse environments.
>
> ***
> We sincerely hope our clarifications above have addressed your questions.

---

> > ### Comment · Reviewer_33rx · 2025-08-05
> >
> > Thank you for your detailed rebuttal and clarifications. I appreciate the effort you’ve put into addressing the points raised in my review.
> >
> > I believe that some points need further clarification, particularly regarding the performance inconsistency and inevitable catastrophic forgetting, which do not show improvement.
> >
> > Nevertheless, in light of the overall contributions, I have decided to raise my score accordingly to "Borderline accept".
> >
> > Good Luck!

---

> > > ### Author Response · Authors · 2025-08-05
> > >
> > > Dear Reviewer,
> > >
> > > Thank you for your constructive feedback. We are delighted that you found our responses satisfactory.
> > >
> > >
> > > Should you have any further questions, we are happy to provide additional clarifications.
> > >
> > > Thank you again for your time and consideration.
> > >
> > > Sincerely,
> > >
> > > The Authors

---

### Official Review · Reviewer_qLPG · 2025-07-02

**Clarity:** 3
**Significance:** 3
**Originality:** 2
**Rating:** 4
**Confidence:** 4

**Summary:**

This paper aims to address catastrophic forgetting and inefficient knowledge utilization in continual reinforcement learning (RL). The authors propose Continual Knowledge Adaptation for Reinforcement Learning (CKA-RL), a continual RL algorithm. CKA-RL maintains a pool of task-specific knowledge (policy’s model parameters) from previously learned tasks and leverages this knowledge to adapt an agent to a new task. To reduce the memory accumulated by historical knowledge, CKA-RL also includes an adaptive knowledge merging mechanism that combines similar knowledge vectors in the pool.

CKA-RL is compared with other methods across Meta-World, SpaceInvaders, and Freeway, and outperforms them in terms of average performance and forward transfer. CKA-RL is also more efficient in terms of memory usage and inference time when evaluated on SpaceInvaders.

**Questions:**

1. Since the model parameters of the first task will be set as the base parameter, does the choice of this first task matter?  In other words, does the sequence of tasks affect the results?
2. (Section 4.2) “For flexibility, we define the null knowledge as…”  What does “flexibility” mean here, and why does defining the null knowledge improve the flexibility?
3. How are betas updated?
4. (Section 4.2) “By setting alpha_1 = 1, the model can disregard previously learned knowledge when it is not beneficial for learning new tasks.”  Why?
5. Do you store the betas learned in different tasks? If not, how do you restore the policies for the tasks that have already been learned?
6. (Table 2) Why does the performance, not only the memory efficiency, improve after adding the adaptive knowledge merging mechanism?
7. (Section 5.4) Why does CKA-RL lead to superior initial performance and rapid convergence?
8. How many random seeds did you run for each experiment?

**Ethical Concerns:**

["NO or VERY MINOR ethics concerns only"]

**Final Justification:**

The authors have provided additional math insights, experiments, and analyses, which help with understanding the underlying mechanisms of their methods. I found the merging mechanism and its results interesting and decided to raise my rating accordingly.

**Limitations:**

Yes

**Quality:**

3

**Strengths And Weaknesses:**

# Strengths
1. The paper is written with a clear flow, and one can easily catch the high-level idea of the proposed method.
2. The idea of including a knowledge merging mechanism is crucial, in my opinion, for scalable continual RL algorithms.
3. The experimental results show that CKA-RL consistently outperforms other evaluated methods in terms of average performance and forward transfer. Some ablation studies are also provided to help understand the effectiveness of the two components in CKA-RL. Especially, I found Figure 11 very interesting, as it demonstrates how distinct each knowledge vector is.

# Weaknesses
1. The motivation and design of CKA-RL are not sufficiently elaborated.
- Knowledge vectors: In Continual RL, various forms of knowledge can be transferred across tasks, such as policies, feature representations, reward functions, and reasoning models. In this paper, the knowledge vectors are specifically designed to be the “residual model parameters” of a policy. It would be easier to follow if the authors could explain why they chose this specific design.
- The normalized adaptation factors, alphas and betas: After reading the paper, I still have a hard time understanding how these parameters are learned. Specifically, how are betas updated?
- The similarity between knowledge vectors: The paper proposed that the normalized cosine similarity of two vectors of residual policy parameters can represent the similarity between two policies. However, I believe there is a difference between the representation and functional similarity of neural networks. Thus, even if the cosine similarity is close to 1, it does not guarantee that the two policies perform similarly.
- Following the previous point, simply merging two knowledge vectors can result in new knowledge with very different behaviors.
2. Ablation studies can be more comprehensive.
- Effects of knowledge pool size: This analysis is conducted only on Freeway, not on other environments. In addition, the size evaluated only ranges between 3 and 7, which is relatively small and may not necessarily generalize to different scales of pool size.
- Reward curve comparison in Figure 4: Which environment is this analysis conducted in?
- Knowledge Retention of CKA-RL: Without statistical significance results, Table 3 does not necessarily indicate that CKA-RL outperforms other baselines in mitigating catastrophic forgetting, as the performance difference is not substantial.
- Memory efficiency and inference efficiency: These analyses are only performed on SpaceInvaders, and the results for other environments are not provided.
3. The statistical significance results are not presented in the paper. Specifically, the number of random seeds used for training is not specified, and most figures and tables do not provide the standard deviation of the results.
4. In the Introduction, the paper mentions that CKA-RL aims to address cross-task conflict; however, after reading the paper, I do not find that CKA-RL can address conflicts between tasks.

---

> ### Author Rebuttal · Authors · 2025-07-31
>
> >Q1. Why did you choose residual model parameters for knowledge vectors?
>
> **A1.** Thank you for this insightful comment regarding our knowledge vector design. Clarify as follows:
> * **Optimal Stability-Plasticity Balance [18].** Unlike full policy parameters exhibiting high cross-task correlation, residual vectors $v_k$ capture task-specific adaptations in **nearly decoupled directions** (Fig. 11a, similarity: −0.24–0.12). This minimizes interference while enabling effective transfer, with $\theta_{base}$ providing stability and $v_k$ enabling plasticity.
> * **Comprehensive Policy Adaptation:** The residual vector compactly encodes all necessary adjustments to the base policy’s behavior, as validated by near-perfect performance on complex tasks.
> * **Scalability Through Merging:** Fig. 5a proves our method maintains **constant memory** (11.49MB after 10 tasks) while CompoNet requires 108.21MB.
>
> >Q2. How are betas updated?
>
> **A2.** Thank you for your insightful comment on the learning mechanism of adaptation factors. The $\beta$ are directly optimized during the training of each new task through standard gradient-based methods. Specifically:
> * For each new task, we initialize a set of learnable parameters $\beta_k=[\beta_1^k,\ldots,\beta_{k-1}^k]$.
> * These $β^k$ parameters are directly optimized alongside the current task knowledge vector $v_k$ during the reinforcement learning process.
> * The gradients for $β^k$ come from the RL objective function, just like gradients for policy parameters.
>
> >Q3. Design of similarity between knowledge vectors.
>
>
> **A3.** Thanks for your comment. Clarify as follows:
> * **Cosine Similarity vs. Functional Similarity.**  Cosine similarity in parameter space does not directly capture the functional similarity between policies. However,  task (or knowledge) vectors [18] are defined as the difference between fine-tuned and base models, exhibit meaningful structure: near-orthogonal vectors often correspond to disjoint tasks, and similar vectors often lead to shared behaviors. While not a perfect proxy, cosine similarity remains a practical measure of representation similarity, which correlates with functional behavior in many settings [18].
> * **Empirical results.** The high cosine similarity between certain knowledge vectors suggests shared structure, and we confirm this with empirical results:
>     * Our merging mechanism preserves performance across tasks, showing no degradation in previously learned policies.
>     * From Fig. 11, the near-orthogonality of learned vectors helps reduce interference, supporting functional retention.
>
> >Q4. Effects of knowledge pool size.
>
> **A4.** Thanks for your insightful comment. **Please see Reviewer-dAWz, A6**.
>
> >Q5. Which environment in Fig. 4?
>
> **A5.**  Fig. 4  is in the **SpaceInavders environment**.
>
> >Q6. The standard forgetting metric.
>
> **A6.** Thanks for your insightful comment. **Please see Reviewer-33rx, A4**.
>
> >Q7. Memory efficiency and inference efficiency.
>
> **A7.** Thanks. We have conducted thorough memory and inference efficiency evaluations on both Meta-World and Freeway environments. From Tabs A&B, CKA-RL maintains nearly constant parameter size and inference latency through vector merging, while other methods show linear growth and increasing computational overhead.
>
> Tab. A:  Analysis in the Meta-World.
> ||Task ID|5|10|50|150|200|300|
> |-|-|-|-|-|-|-|-|
> |Parameter Memory (MB)|ProgNet|5.64|17.22|334.79|2878.74|5088.21|11382.16|
> ||CompoNet|7.18|14.43|72.48|217.60|290.17|435.29|
> ||CKA-RL|5.64|10.34|10.34|10.34|10.34|10.34|
> |Inference latency (s)|ProgNet|0.0010|0.0021|0.0315|0.2558|0.4610|1.0353|
> ||CompoNet|0.0023|0.0045|0.0213|0.0636|0.0842|0.1264|
> ||CKA-RL|0.0002|0.0001|0.0002|0.0003|0.0002|0.0002|
>
> Tab. B: Analysis in the Freeway.
> ||Task ID|0|1|3|5|7|
> |-|-|-|-|-|-|-|
> |Parameter Memory (MB)|ProgNet|7.44|13.82|26.61|39.39|52.17|
> ||CompoNet|13.86|24.35|45.31|66.28|87.25|
> ||CKA-RL|7.44|8.45|10.48|12.51|14.54|
> |Inference latency (s)|ProgNet|0.0018|0.0017|0.0027|0.0041|0.0061|
> ||CompoNet|0.0015|0.0019|0.0032|0.0046|0.0035|
> ||CKA-RL|0.0015|0.0015|0.0015|0.0015|0.0015|
>
>
> >Q8. The statistical significance results.
>
> **A8.** Thanks for your valuable feedback. **Please see Reviewer-Mgpw, A1**.
>
>
> >Q9. Why can CKA-RL address conflicts between tasks?
>
> **A9.** Thanks. Cross-task conflict in CRL primarily manifests as parameter overlap and interference. CKA-RL mitigates this through the knowledge vector.
> * **Orthogonal knowledge vectors.** From Fig. 11a, CKA-RL’s task-specific knowledge vectors exhibit near-orthogonal structures in parameter space. This minimizes representational overlap between task-specific adaptations, directly reducing the potential for conflict when learning new tasks.
> * **Stable base parameters.** The fixed base parameter preserves general knowledge across tasks, ensuring that foundational abilities are not overwritten by task-specific updates. This separation prevents new task parameters from disrupting core knowledge needed for earlier tasks.
> * **Adaptive knowledge weighting.** $\alpha_k$ dynamically adjusts the contribution of historical knowledge vectors to new tasks, allowing the model to prioritize relevant historical knowledge while suppressing conflicting information.
>
> >Q10. Task Sequence Impact.
>
> **A10.** Thanks for your comment. Clarify as follows:
> * **Task Sequence Impact.** The choice of the first task as base parameter does matter. Results confirm that different task sequences lead to performance variations (about 3.7% difference in performance).
> * **Robustness Analysis.** Despite this sensitivity, CKA-RL demonstrates greater robustness to task ordering than methods like CompoNet, as shown in Tab. C.
>
> Tab. C: Different task orders on the Freeway.
> |Order|METHOD| FREEWAY PERF. |FREEWAY FWT.|
> |-|-|:-:|:-:|
> |Standard|CompoNet|0.7629$\pm$ 0.12|0.7115$\pm$ 0.10|
> ||CKA-RL|**0.7923$\pm$ 0.10**|**0.7429$\pm$ 0.07**|
> |Shuffle|CompoNet|0.7066$\pm$ 0.19|0.6515$\pm$ 0.18|
> ||CKA-RL|**0.8294$\pm$ 0.11**|**0.7704$\pm$ 0.10**|
>
>
> >Q11. Why does defining the null knowledge improve the flexibility?
>
> **A11.** Thanks. The term "flexibility" **refers to the unified mathematical formulation** of knowledge adaptation across all tasks, including the first task $\tau_1$. By defining $v_1 =\theta_1 - \theta_{base} = 0$ and including it in the knowledge pool $\mathcal{V}$. In Eqn.(4), the adaptation factors $\alpha_j^k$are derived from learnable parameters $\beta^k$ through softmax. When $v_1 = 0$, the value of $\alpha_1^k$ becomes irrelevant to the final policy parameter, **allowing the model to "learn" to ignore historical knowledge when it's not beneficial.**
>
>
> >Q12. “The model can disregard previously learned knowledge when it is not beneficial for learning new tasks.” Why?
>
> **A12.** Thanks. This statement refers to the role of the null knowledge vector $v_1$ in our framework.
> * $\alpha_1$ is the adaptation factor corresponding to $v_1$, the null knowledge vector. Since the sum of all $\alpha^k_j$ (j = 1, ..., k-1) is constrained to 1, setting $\alpha_1 = 1$ forces all other factors $\alpha^k_2, ..., \alpha^k_{k-1}$ to be 0.
> * Because $v_1 = 0, \alpha_1 = 1$ means the model’s parameter update for the new task $\tau_k$ depends **only** on the base parameter and the current task’s knowledge vector, with no contribution from other historical knowledge vectors $v_2, ..., v_{k-1}$. In this case, the model effectively disregards previously learned knowledge stored in these vectors. This design ensures the model avoids over-reliance on unhelpful historical knowledge.
>
> >Q13. Do you store learned betas and how are policies restored?
>
> **A13.** Thanks. We do not store the $\beta^k$ parameters.
> * **Role of $\beta^k$ Parameters.** The $\beta^k$ serve as temporary, task-specific variables during the training of a new task. Their role is to dynamically determine the adaptation factors $\alpha^k$ that weight historical knowledge vectors, facilitating the acquisition of the current task’s knowledge vector.
> * **Restoring Policies for Previously Learned Tasks.** To restore the policy for a previously learned task $\tau_i$, we rely on the stored knowledge vector $v_i$ and the fixed base parameter $\theta_{base}$. The policy parameter is defined as: $\theta_i = \theta_{base} + v_i$.  Since $v_i$ encapsulates all task-specific adaptations needed for $\tau_i$, restoring the policy only requires retrieving $v_i$ from the knowledge pool and combining it with $\theta_{base}$.
>
> Q14. Why does performance improve with the adaptive knowledge merging mechanism?
>
> **A14.** Thanks. Clarify as follows:
> * **Reducing interference from conflicting knowledge.** By merging redundant or conflicting vectors, the model avoids carrying irrelevant or contradictory information into new task learning. This minimizes "noise" in knowledge transfer, allowing the model to focus on truly useful historical insights.
> * **Preserving and amplifying critical information.** The merging process retains essential task-specific knowledge while consolidating similar vectors into more generalized representations. This strengthens the signal of transferable knowledge, enabling more effective adaptation to new tasks.
>
> >Q15. Why does CKA-RL lead to superior initial performance and rapid convergence?
>
> **A15.** Thanks. The reason is that CKA-RL ability to leverage high-quality historical knowledge as a useful prior while minimizing interference during new task learning.
> * **Effective reuse of historical knowledge.** CKA-RL initializes the current task’s knowledge vector to 0 and dynamically weights relevant historical vectors via $\alpha^k$, directly boosting initial performance.
> * **Refined knowledge signals from merging.** The adaptive knowledge merging mechanism consolidates redundant vectors, strengthening the signal of transferable knowledge. This ensures the historical knowledge used to initialize new tasks is more relevant and less noisy, further speeding up learning.

---

> > ### Comment · Reviewer_qLPG · 2025-08-04
> >
> > Thank you for your detailed responses. The added explanations and experiments help clarify some of my questions and concerns.
> >
> > I have read through all other reviews and authors’ responses. After careful consideration, I will maintain my current rating, mainly because (1) the improved performance and the proposed method does not necessary correlate without detailed analyses or mathematical studies (currently, it mainly comes from authors’ explanations) and (2) the novelty and technical contributions of the proposed methods requires more validation.
> >
> > I will keep following the discussions and adjust my rating if I find suitable.

---

> > > ### Author Response · Authors · 2025-08-05
> > > **Detailed analyses and mathematical studies**
> > >
> > > We sincerely appreciate your time and constructive feedback throughout the review process. We also thank the reviewer for recognizing that the knowledge merging mechanism is crucial for scalable continual RL.
> > >
> > > >Q1. The improved performance and the proposed method does not necessary correlate without detailed analyses or mathematical studies.
> > >
> > > **A1.** We appreciate this opportunity to provide rigorous theoretical analysis and additional experimental evidence.
> > > * **Preliminaries (from the paper).** We parameterize the policy for task $\tau_k$ as $\theta_k \;=\; \theta_{\text{base}} \;+\; \sum_{j=1}^{k-1}\alpha^k_j\, v_j \;+\; v_k,
> > > \quad \text{with } \sum_{j=1}^{k-1}\alpha^k_j=1,$ where the normalized adaptation factors are produced by a softmax over learnable $\beta^k$ (Eq. (3)–(4)). When the pool size exceeds $K_{\max}$, we merge the most similar knowledge vectors by averaging (Eq. (5)–(7)).
> > > * **Lemma 1 (Drift bound under convex reuse).** Let $\Delta_k := \theta_k-\theta_{\text{base}}=\sum_{j<k}\alpha^k_j v_j + v_k$. Because $\alpha^k$ are nonnegative and sum to 1, $\|\Delta_k\| \;\le\; \Big\|\sum_{j<k}\alpha^k_j v_j\Big\| + \|v_k\|\;\le\; \sum_{j<k}\alpha^k_j\|v_j\| + \|v_k\|\;\le\; \max_{j<k}\|v_j\| + \|v_k\|.$ Hence the deviation from the base is controlled by magnitudes of (a small subset of) vectors actually reused and the current task vector. This uses the normalization of $\alpha^k$ given by Eq. (3)–(4).
> > >     * **Corollary 1 (Lipschitz performance stability).** If the task-$k$ return $J_k(\theta)$ is $L$-Lipschitz in parameters, then $|J_k(\theta_k)-J_k(\theta_{\text{base}}+v_k)|\;\le\; L\,\big\|\sum_{j<k}\alpha^k_j v_j\big\|\;\le\; L \sum_{j<k}\alpha^k_j\|v_j\|.$ Thus, reusing historical vectors cannot hurt beyond a tunable, data-dependent bound, and the bound tightens as $\alpha^k$ concentrates on small-norm or well-aligned vectors (see Eq. (3)–(4)).
> > > * **Lemma 2 (Interference reduces with near-orthogonality).** Let $S_{ij}=\frac{v_i^\top v_j}{\|v_i\|\|v_j\|}$ be the cosine similarity. If $|S_{ij}|\le \varepsilon \ll 1$ for $i\neq j$, then $\Big\|\sum_{j<k}\alpha^k_j v_j\Big\|^2= \sum_{j<k}(\alpha^k_j)^2\|v_j\|^2 + \sum_{i\neq j}\alpha^k_i\alpha^k_j \|v_i\|\,\|v_j\| S_{ij} \;\le\; \sum_{j<k}(\alpha^k_j)^2\|v_j\|^2 + \varepsilon\!\sum_{i\neq j}\alpha^k_i\alpha^k_j \|v_i\|\,\|v_j\|.$ Hence the “cross-task” term is $O(\varepsilon)$. Our empirical cosine analysis shows knowledge vectors are nearly orthogonal, with off-diagonal values in $[-0.24,0.12]$, while full fine-tuned parameters have strong correlations ($0.93\sim 0.95$). **This matches the design goal of reducing interference.**
> > >     * **Corollary 2.** Combining Lemma 1–2, both the drift and the cross-terms that cause interference are controlled—explaining the improved retention seen in the final-policy evaluation (Table 3 in the paper).
> > > * **Lemma 3 (Bounded error of adaptive merging).** Suppose we must replace $(v_m,v_n)$ by $v_{\text{merge}}=\tfrac{1}{2}(v_m+v_n)$ when $|V|>K_{\max}$. For any convex coefficients $\lambda,\mu\ge 0,\lambda+\mu=1$, $\big\|\lambda v_m+\mu v_n - v_{\text{merge}}\big\|\;\le\; \tfrac{1}{2}\|v_m-v_n\|.$ If their cosine similarity $S_{mn}\ge 1-\delta$ with $\delta\in[0,2]$, then $\|v_m-v_n\|\le \sqrt{2(1-S_{mn})}\,\max(\|v_m\|,\|v_n\|)\le \sqrt{2\delta}\,\max(\|v_m\|,\|v_n\|)$. Thus the parameter perturbation induced by merging is $O(\sqrt{\delta})$.
> > >     * **Corollary 3 (Performance stability under merging).** With $L$-Lipschitz $J_k$,$|J_k(\theta^{\text{after-merge}})-J_k(\theta^{\text{before-merge}})|\;\le\; \tfrac{L}{2}\|v_m-v_n\|
> > > \;\le\; \tfrac{L}{2}\sqrt{2(1-S_{mn})}\,\max(\|v_m\|,\|v_n\|).$ Hence merging similar vectors (large $S_{mn}$) has a small, explicitly bounded effect, **justifying our “merge-the-most-similar” rule in Eq. (6)–(7)**.
> > > * **More Experimental Analysis.** To further validate the effectiveness of CKA-RL and provide deeper insights into its working mechanisms, we conducted additional experiments that complement our main findings. As shown in Tab D, simply averaging knowledge vectors without adaptive weighting performs worse than the baseline (0.7437 performance), **confirming that naive merging without adaptation actually harms performance.** Adding only the Continual Knowledge Adaptation mechanism improves performance to 0.7821 and forward transfer to 0.7321, demonstrating the value of dynamic knowledge weighting.
> > >
> > >
> > > Tab. D： Effectiveness of components in CKA-RL in Freeway task sequence.
> > > | Method           | Perf.   | Fwt.    |
> > > |------------------|---------|---------|
> > > | Base             | 0.7532  | 0.6935  |
> > > | Base+Adapt       | 0.7821  | 0.7321  |
> > > | Base+Avg         | 0.7437  | 0.7127  |
> > > | Base+Adapt+Merge | **0.7923** | **0.7429** |

---

> > > ### Author Response · Authors · 2025-08-05
> > > **The novelty and technical contributions**
> > >
> > > >Q2. The novelty and technical contributions of the proposed methods require more validation.
> > >
> > > **A2.** Thank you for your valuable feedback regarding the novelty and technical contributions of our proposed method. We appreciate the opportunity to provide more comprehensive validation of CKA-RL's innovations. **We believe the novelty of our work lies in addressing the unique challenges of Continual Reinforcement Learning (CRL)**, which fundamentally differ from supervised continual learning settings.
> > > * **CKA-RL as a CRL-Specific Solution.** The fundamental novelty of CKA-RL is its explicit design for the distinctive challenges of continual reinforcement learning (CRL), in which agents must continuously adapt to non-stationary environments [r1,r2]. Because CRL differs substantially from supervised continual learning, approaches that succeed in the supervised setting often fail when directly applied to CRL [r3].
> > > * **CKA-RL's Solutions to CRL-Specific Challenges.**
> > >     * **Continual Knowledge Adaptation Mechanism.** **CKA-RL** introduces a dynamic knowledge adaptation mechanism, where task-specific knowledge vectors are selectively weighted using learnable adaptation factors. This allows our model to **dynamically adjust the relevance of historical knowledge** based on the current task, which is crucial in reinforcement learning environments where task relationships are complex and evolving. As demonstrated in Figure 4 and Tab. E, this leads to significantly higher initial rewards at the beginning of each new task.
> > >     * **Adaptive Knowledge Merging.**  CKA-RL incorporates an adaptive merging strategy and a fixed-size knowledge vector pool, **ensuring efficient memory usage while also addressing the interference between similar tasks**. Our fixed-size pool approach effectively limits parameter growth while avoiding the noise from conflicting vectors, which can impair learning efficiency. This aspect is particularly crucial in reinforcement learning tasks, where the agent must continuously interact with the environment and adapt its policy without overwhelming its memory with redundant or irrelevant information. Figure 5a demonstrates that CKA-RL maintains constant parameter memory beyond the fifth task while preserving sufficient policy diversity for exploration—a capability impossible with methods that linearly accumulate task modules. This is not merely a resource constraint but an essential component for maintaining policy diversity in non-stationary environments.
> > >
> > > Tab. E Return Comparison (at 6000, 7000, 8000, 9000, 10000 steps)
> > >
> > > | Method | 100k | 200k | 300k | 400k | 500k | 600k | 700k | 800k | 900k | 1000k |
> > > | --- | --- | --- | --- | --- | --- | --- | --- | --- | --- | --- |
> > > | ProgNet | 296.2 $\pm$ 210.6 | 322.2 $\pm$ 215.9 | 323.5 $\pm$ 119.5 | 304.1 $\pm$ 147.0 | 375.4 $\pm$ 149.1 | 397.5 $\pm$ 159.1 | 498.9 $\pm$ 123.8 | 519.0 $\pm$ 124.9 | 485.0 $\pm$ 139.1 | 310.0 $\pm$ 5.0 |
> > > | CbpNet | 425.8 $\pm$ 206.3 | 368.3 $\pm$ 206.7 | 341.7 $\pm$ 209.2 | 375.8 $\pm$ 246.1 | 329.5 $\pm$ 74.1 | 417.5 $\pm$ 83.8 | 416.7 $\pm$ 166.0 | 482.0 $\pm$ 173.3 | 507.7 $\pm$ 121.3 | 445.0 $\pm$ 0.0 |
> > > | CompoNet | 363.1 $\pm$ 132.8 | **517.2 $\pm$ 98.2** | 439.5 $\pm$ 178.2 | 472.0 $\pm$ 170.6 | 440.5 $\pm$ 159.0 | **450.6 $\pm$ 106.6** | 437.1 $\pm$ 134.7 | 564.1 $\pm$ 149.8 | 486.1 $\pm$ 139.2 | 391.7 $\pm$ 155.8 |
> > > | CKA-RL | **506.9 $\pm$ 157.1**| 434.1 $\pm$ 135.5 | **471.7 $\pm$ 118.9** | **480.0 $\pm$ 134.2** | **546.7 $\pm$ 142.6** | 400.5 $\pm$ 117.8 | **550.6 $\pm$ 179.4** | **579.0 $\pm$ 203.0** | **527.1 $\pm$ 220.8**| **493.9 $\pm$ 138.4** |
> > >
> > >
> > > [r1] Loss of plasticity in deep continual learning, Nature 2024.
> > >
> > > [r2] Towards continual reinforcement learning: A review and perspectives, JAIR 2022.
> > >
> > > [r3] Replay-enhanced Continual Reinforcement Learning, TMLR 2023.
> > >
> > > ***
> > > We hope the above clarifications address your concerns. We aim to resolve all remaining issues to enhance the clarity and rigor of the manuscript.

---

> > > ### Author Response · Authors · 2025-08-06
> > >
> > > Dear Reviewer,
> > >
> > > Thank you for your constructive feedback and for acknowledging that our rebuttal has addressed most of your concerns. We are delighted that you found our responses satisfactory. We also appreciate your initial assessment highlighting that the paper is "written with a clear flow," that "including a knowledge merging mechanism is crucial… for scalable continual RL,” and that CKA-RL “consistently outperforms other evaluated methods… in average performance and forward transfer," with Figure 11 "demonstrat[ing] how distinct each knowledge vector is."
> > >
> > >
> > > We would like to take this opportunity to reiterate the core contributions of our work, which we believe make a significant contribution to the field:
> > >
> > > * **Continual Knowledge Adaptation.** We introduce learnable, task-conditioned adaptation factors that selectively weight historical knowledge, enabling rapid and stable adaptation in non-stationary RL environments.
> > > * **Task-Specific Knowledge Vectors.** We decouple task-specific updates from a stable base policy via compact knowledge vectors, reducing policy interference and mitigating catastrophic forgetting while preserving prior competence.
> > > * **Adaptive Knowledge Merging with a Bounded Pool.** We propose similarity-aware consolidation that maintains a fixed-size pool of salient knowledge, **controlling memory growth and inference latency** while limiting conflicts among related tasks.
> > > * **Scalability and Early-Stage Learning Efficiency.** Experiments across standard CRL benchmarks (and extended task sequences) show faster early adaptation and strong long-horizon performance under tight memory budgets, supported by multi-seed statistics and comprehensive ablations.
> > >
> > > We hope that these innovations and the substantial empirical gains will merit a score improvement. Should you have any further questions, we are happy to provide additional clarifications.
> > >
> > > Thank you again for your time and consideration.
> > >
> > > Sincerely,
> > >
> > > The Authors

---

> > > > ### Comment · Reviewer_qLPG · 2025-08-06
> > > >
> > > > Thank you for the additional analyses, experiments, and mathematical insights. I think these help a lot with understanding the underlying mechanisms and your contributions. I decided to raise my rating accordingly.

---

> > > > > ### Author Response · Authors · 2025-08-06
> > > > >
> > > > > Dear Reviewer,
> > > > >
> > > > > Thank you for your constructive feedback. We are delighted that you found our responses satisfactory.
> > > > >
> > > > >
> > > > > Should you have any further questions, we are happy to provide additional clarifications.
> > > > >
> > > > > Thank you again for your time and consideration.
> > > > >
> > > > > Sincerely,
> > > > >
> > > > > The Authors

---

### Official Review · Reviewer_Mgpw · 2025-07-03

**Clarity:** 3
**Significance:** 2
**Originality:** 2
**Rating:** 4
**Confidence:** 3

**Summary:**

This paper introduces Continual Knowledge Adaptation for Reinforcement Learning (CKA-RL), a new method for continual reinforcement learning (CRL) designed to mitigate catastrophic forgetting and improve knowledge transfer while maintaining scalability. The core idea is to represent task-specific knowledge as "knowledge vectors." To address the challenge of ever-growing memory costs, the authors propose an "Adaptive Knowledge Merging" mechanism, which identifies and merges the most similar pairs of knowledge vectors in a fixed-size pool, thus keeping the memory footprint bounded.

**Questions:**

For related work, could the authors also discuss more recent crl specific papers like 1. Prediction and Control in Continual Reinforcement Learning. Anand et al. 2. Knowledge Retention for Continual Model-Based Reinforcement Learning. Sun et al. etc. ?

**Ethical Concerns:**

["NO or VERY MINOR ethics concerns only"]

**Limitations:**

yes

**Quality:**

2

**Strengths And Weaknesses:**

Strengths:
1. Important problem and the algorithm generally makes sense.
2. The analysis in Figure 5 and Appendix D clearly demonstrates the superior memory and inference efficiency of CKA-RL.

Weaknesses:
1. Significance of Empirical Results is Unclear: My biggest concern is the lack of statistical rigor in the experimental validation. The performance improvements reported in the main results (Table 1) appear to be quite marginal (e.g., ~1-4% in many cases). The authors state in the checklist that results are from a single run with a fixed seed and do not report standard deviations. For an empirical RL paper, this is a critical omission. Without error bars over multiple runs, it is impossible to determine if these small advantages are statistically significant or simply the result of run-to-run variance.

2. Moderate Methodological Novelty: The core idea of decomposing model updates into task-specific vectors is heavily inspired by prior work in model editing and task arithmetic. While its application to CRL is new, the fundamental mechanism is not

---

> ### Author Rebuttal · Authors · 2025-07-31
>
> We are deeply grateful to you for recognizing the strengths of our work, particularly the clear presentation of our methodology and the balanced trade-off CKA-RL achieves among plasticity, stability, and scalability. We sincerely appreciate your acknowledgment of our comprehensive experimental validation and the potential impact of our approach on the continual reinforcement learning community.
>
> >Q1. Significance of Empirical Results is Unclear: My biggest concern is the lack of statistical rigor in the experimental validation. The performance improvements reported in the main results (Table 1) appear to be quite marginal (e.g., ~1-4% in many cases). The authors state in the checklist that results are from a single run with a fixed seed and do not report standard deviations. For an empirical RL paper, this is a critical omission. Without error bars over multiple runs, it is impossible to determine if these small advantages are statistically significant or simply the result of run-to-run variance.
>
> **A1.** Thank you for your valuable comment regarding the statistical rigor of our experimental evaluation. We will address this in the revised version, as detailed below:
> * **Multiple Random Seeds Used.** Our experiments were conducted with 10 different random seeds, consistent with standard practice in the recent CRL method CompoNet. We acknowledge that this detail was not explicitly stated in our initial submission, which was an oversight.
> * **Statistical Significance Evidence.** We have computed and included standard deviations for all reported results. The updated results table (provided in Tab. A) now shows mean performance ± standard deviation across all task sequences and methods. Notably, CKA-RL consistently demonstrates lower variance compared to baselines.
> * **Contextualizing the Magnitude of Improvements.** While the percentage improvements may appear modest numerically, they represent meaningful advances in the context of continual RL. 1) The field of CRL has seen diminishing returns in performance gains, with recent SOTA methods (CompoNet, CbpNet) showing only average 1-3% improvements over prior work (CReLUs); 2) Our method achieves these gains while using significantly less memory (about 5.8MB vs CompoNet's about 108.21MB at the 10th task).
>
>
> Tab. A: Summary of results across all task sequences and methods.
> |METHOD|META-WORLD PERF. |META-WORLD FWT.| SPACEINVADERS PERF.|SPACEINVADERS FWT.| FREEWAY PERF. |FREEWAY FWT.|
> |-|:-:|:-:|:-:|:-:|:-:|:-:|
> |ProgNet|0.4157$\pm$ 0.49|-0.0379$\pm$ 0.13 |0.3757$\pm$ 0.27|-0.0075$\pm$ 0.22|0.3125$\pm$ 0.27 |0.1938$\pm$ 0.31|
> |PackNet|0.2523$\pm$ 0.40|-0.6721$\pm$ 1.40|0.2299$\pm$ 0.30|-0.075$\pm$ 0.13|0.2767$\pm$ 0.36| 0.1970$\pm$ 0.32|
> |MaskNet|0.3263$\pm$ 0.47 |-0.3695$\pm$ 0.45|0$\pm$ 0 |-0.3866$\pm$ 0.53|0.0644$\pm$ 0.17|-0.0503$\pm$ 0.12|
> |CReLUs|0.3789$\pm$ 0.47|-0.0089$\pm$ 0.24|0.8873$\pm$ 0.10|0.5308$\pm$ 0.29|0.7835$\pm$ 0.13|0.7303$\pm$ 0.12|
> |CompoNet|0.4131$\pm$ 0.50|-0.0055$\pm$ 0.20|0.9828$\pm$ 0.02|0.6963$\pm$ 0.32|0.7629$\pm$ 0.12|0.7115$\pm$ 0.10|
> |CbpNet|0.4368$\pm$ 0.50 |-0.0826$\pm$ 0.22|0.8392$\pm$ 0.11|0.4844$\pm$ 0.28|0.7678$\pm$ 0.10|0.7201$\pm$ 0.07|
> |CKA-RL|**0.4642 $\pm$ 0.50** |**-0.0032$\pm$ 0.21**|**0.9928 $\pm$ 0.01**|**0.7749$\pm$ 0.20**|**0.7923$\pm$ 0.10**|**0.7429$\pm$ 0.07**|
>
>
> >Q2. Moderate Methodological Novelty: The core idea of decomposing model updates into task-specific vectors is heavily inspired by prior work in model editing and task arithmetic. While its application to CRL is new, the fundamental mechanism is not.
>
> **A2.** Thank you for your thoughtful comment regarding the methodological novelty of our work. We would like to clarify the substantive contributions of CKA-RL beyond its conceptual inspiration from model editing techniques.
> * **Acknowledgment of Inspiration.**  You are correct that our knowledge vector concept is inspired by prior work in model editing and task arithmetic, as explicitly acknowledged in Section 4.1 of our paper.
> * **Fundamental Novelty in CRL Context.**  The core contribution of CKA-RL lies not merely in applying this concept to continual reinforcement learning, but in developing a complete framework specifically designed for the unique challenges of CRL.
>     * **Dynamic Knowledge Adaptation Mechanism** Unlike static model merging approaches in supervised learning that typically combine pre-trained models with fixed weights, our Continual Knowledge Adaptation strategy dynamically determines how to utilize historical knowledge through learnable adaptation factors. This allows the agent to flexibly select and weight relevant knowledge based on the current task, which is critical in non-stationary RL environments where task relationships are complex and evolving.
>     * **Adaptive Knowledge Merging for CRL.** Our merging mechanism (Eqn. (8)) is specifically designed for the sequential nature of continual RL, where tasks arrive sequentially rather than simultaneously. The similarity metric and merging strategy we propose are tailored to the unique characteristics of RL knowledge transfer, considering both parameter similarity and performance impact.
>
>
> >Q3. For related work, could the authors also discuss more recent crl specific papers like 1. Prediction and Control in Continual Reinforcement Learning. Anand et al. 2. Knowledge Retention for Continual Model-Based Reinforcement Learning. Sun et al. etc. ?
>
> **A3.** Thank you for your valuable suggestion regarding the inclusion of more recent CRL-specific papers in our related work section. We appreciate this insightful observation and will incorporate these important works in our revision. Below is our detailed response:
> * **Inclusion of Suggested Works.** We sincerely thank you for bringing "Prediction and Control in Continual Reinforcement Learning" by Anand et al. and "Knowledge Retention for Continual Model-Based Reinforcement Learning" by Sun et al. to our attention. These represent significant recent contributions to the CRL field that were inadvertently omitted from our initial submission.
> * **Comparison with Anand et al.'s Work.** Anand et al. propose decomposing the value function into permanent and transient components that update at different timescales. While their approach shares conceptual similarities with our work in addressing the stability-plasticity dilemma, there are fundamental differences: Anand et al. focus on value function estimation, while CKA-RL operates directly on policy parameters. Our knowledge vector approach decomposes policy parameters into stable base weights ($θ_{base}$) and task-specific knowledge vectors ($v_i$), rather than manipulating value functions.
> * **Relationship to Sun et al.'s DRAGO.** Sun et al.'s DRAGO framework introduces innovative mechanisms for knowledge retention in model-based CRL. While highly relevant, our approach differs in scope aspects: DRAGO specifically targets model-based RL where tasks share state spaces and dynamics but differ in rewards, while CKA-RL is designed for both model-free and model-based settings with broader applicability across diverse task distributions.
>
> ***
> We sincerely hope our clarifications above have addressed your concerns. We would be grateful if you could kindly reconsider the evaluation of our paper.

---

> > ### Comment · Reviewer_Mgpw · 2025-08-06
> >
> > Thank you for the response. Most of my concerns are solved. I will remain my current score.

---

> > > ### Author Response · Authors · 2025-08-07
> > >
> > > Dear Reviewer,
> > >
> > > Thank you for your constructive feedback. We are delighted that you found our responses satisfactory.
> > >
> > >
> > > Should you have any further questions, we are happy to provide additional clarifications.
> > >
> > > Thank you again for your time and consideration.
> > >
> > > Sincerely,
> > >
> > > The Authors

---

### Note · Authors · 2025-08-14

Dear ACs, SACs, and PCs,

We sincerely appreciate all reviewers’ time and efforts in reviewing our paper and for the constructive feedback. We are glad that the reviewers appreciate and recognize our contributions: **1) Novel Methodology** (dAWz, qLPG); **2) Balanced Plasticity/Stability/Scalability** (dAWz); **3) Superior Efficiency & Performance** (Mgpw, 33rx, dAWz, qLPG); **4) Key Evidence in Figures** (Mgpw: Fig5/App. D; qLPG: Fig11); **5) Clarity & Reproducibility** (33rx, dAWz, qLPG).

After the rebuttal, we are pleased that **three reviewers (33rx, dAWz, qLPG) raised their scores** to the positive range, while **one reviewer (Mgpw) maintained a positive score**.

We would like to take this opportunity to reiterate the core contributions of our work, which we believe make a significant contribution to the field:

* **Continual Knowledge Adaptation.** We introduce learnable, task-conditioned adaptation factors that selectively weight historical knowledge, enabling rapid and stable adaptation in non-stationary RL environments.
* **Task-Specific Knowledge Vectors.** We decouple task-specific updates from a stable base policy via compact knowledge vectors, reducing policy interference and mitigating catastrophic forgetting while preserving prior competence.
* **Adaptive Knowledge Merging with a Bounded Pool.** We propose similarity-aware consolidation that maintains a fixed-size pool of salient knowledge, **controlling memory growth and inference latency** while limiting conflicts among related tasks.
* **Scalability and Early-Stage Learning Efficiency.** Experiments across standard CRL benchmarks (and extended task sequences) show faster early adaptation and strong long-horizon performance under tight memory budgets, supported by multi-seed statistics and comprehensive ablations.

Thank you for your time and guidance in facilitating this discussion.

Best,

Authors

---

### Decision · Program_Chairs · 2025-09-17

**Decision:**

Accept (poster)

**Comment:**

Summary: This paper addresses the important problem of continual reinforcement learning in non-stationary environments, proposing CKA-RL, which combines dynamic weighting of prior task knowledge, task-specific residual vectors, and an adaptive merging mechanism to maintain bounded memory.

Strengths: The reviewers agree that the paper is clearly written, empirically well-supported, and tackles a relevant challenge. Strengths include the memory/inference efficiency analysis, consistent performance gains across Meta-World and Atari, and the introduction of a merging mechanism that enables scalability.

Weaknesses: The main concerns of the reviewers center on the perceived incremental novelty of task-vector based methods, limited improvement on catastrophic forgetting, and questions about the fairness of comparing bounded-pool scalability to baselines with growing memory.

Rebuttal: The rebuttal successfully clarified questions on experimental results (e.g. adding error bars), provided theoretical analysis (drift bounds, orthogonality, merging error), and added long-horizon experiments, which led three reviewers to raise their scores and the fourth to maintain a positive borderline-accept rating.

Recommendation: Borderline Accept. While this is an incremental contribution, the work represents a step forward in scalable continual RL under bounded resources, with contributions that will interest the community.